# Learning from Adversity: Semantic-Aware Mask Refinement through Adversarial Perturbation

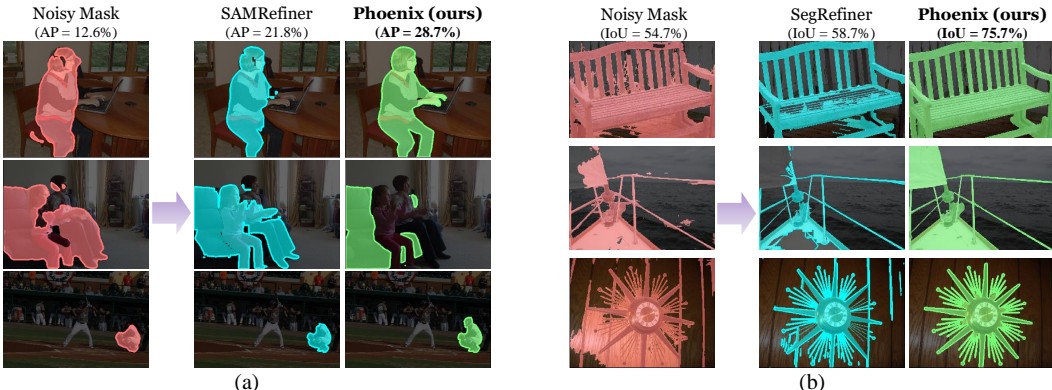

Figure 1: **Qualitative Comparison** of our mask refinement method with existing approaches. The (a) instance segmentation and (b) fine-grained segmentation examples show our superior mask refinement performance on complex structures and boundary details.

## Abstract

Despite significant advances in image segmentation, even state-of-the-art models produce masks with imperfect boundaries, semantic inconsistencies, and structural errors. Mask refinement addresses these limitations, yet current approaches rely on simplistic synthetic noise that fails to capture the complex error patterns of real segmentation models. We introduce Phoenix, a novel framework that leverages adversarial learning to generate semantically meaningful noise patterns and contrastive learning to model refinement relationships. Our approach consists of two key innovations: (1) Adversarial Mask Perturbation, which employs embedding attacks to create semantic-aware noise that mimics real segmentation errors, and (2) Contrastive Mask Refinement Learning, which establishes a tri-directional framework that ensures feature consistency within semantic regions while maintaining separation between classes. Experiments demonstrate that Phoenix significantly outperforms existing methods across diverse tasks, while consistently enhancing state-of-the-art segmentation models with substantial improvements.

## 1 Introduction

Image segmentation provides pixel-level understanding crucial for numerous applications. Despite remarkable progress in segmentation architectures, even state-of-the-art models Cheng et al. (2022); Kirillov et al. (2023); Xie et al. (2021a) exhibit persistent limitations: imprecise boundaries at complex contours, semantic confusion between similar objects, and structural inconsistencies that violate object integrity. These errors stem from fundamental challenges that remain despite extensive model scaling and data collection efforts.

This circumstance establishes mask refinement as a distinct and versatile complementary approach to conventional segmentation methods, directly addressing their systematic limitations. It enhances model performance without architectural changes, making it valuable in cases where model updates are infeasible, such as due to proprietary restrictions, computational limitations, or data

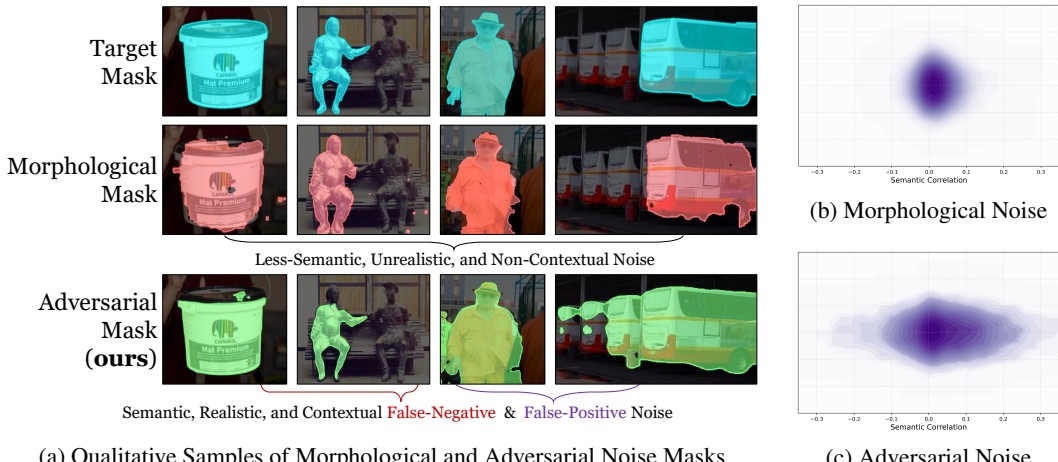

(a) Qualitative Samples of Morphological and Adversarial Noise Masks

(b) Morphological Noise

(c) Adversarial Noise

Figure 2: **Qualitative Comparison** of noise patterns (a) between morphological and our adversarial noise masks. (b) Semantic correlation distribution of morphological noise showing a narrow distribution, and (c) our adversarial noise showing a broader distribution and more diverse error types.

collection costs. Furthermore, mask refinement is pivotal in label-efficient learning, including semi-supervised Kim et al. (2023); Wang et al. (2022) and weakly-supervised settings Kim et al. (2022); Tian et al. (2021), transforming low-quality pseudo-labels into reliable supervision, which maximizes learning outcomes even with limited annotations.

The construction of realistic noisy and clean mask pairs is central to effective refinement learning. Recent efforts Lin et al. (2025); Tang et al. (2021) have advanced mask refinement through various approaches, yet significant limitations persist. Methods like SegFix Yuan et al. (2020) and SegRefiner Wang et al. (2023) rely on synthetic noise generated through morphological operations, producing simplistic, spatially random perturbations that often fail to capture the structured, context-dependent errors of real neural networks. This fundamental limitation restricts their ability to address the complex challenges encountered in real-world segmentation tasks, as shown in Figure 1.

"*What doesn't kill you makes you stronger.*" This wisdom reflects how adversity can become a catalyst for growth, a principle we translate from philosophy to algorithm design. In natural systems, adaptation occurs in response to meaningful challenges. Similarly, the effective learning system emerges from facing realistic, meaningful obstacles rather than artificial ones. Drawing inspiration from this, we introduce a novel framework for semantic-aware mask refinement through adversarial perturbation. We dub our framework *Phoenix* since it transforms challenging noise patterns into superior refinement capabilities, mirroring how the mythical bird rises stronger from the ashes.

The Phoenix framework builds upon two key innovations. First, Adversarial Mask Perturbation (AMP) employs adversarial embedding attacks to generate semantically meaningful, contextually aware noise patterns. By optimizing against the model's learned representations, AMP creates perturbations that concentrate precisely where real segmentation models struggle most, such as at semantically challenging boundaries and ambiguous regions, as shown in Figure 2. Notably, this approach offers controllability over noise generation patterns while maintaining computational efficiency. Second, our Contrastive Mask Refinement Learning (CMRL) explicitly models the relationships between ground truth, noisy input, and current prediction. Unlike conventional pixel-wise objective functions, this tri-directional contrastive mechanism promotes clear separation between foreground and background features while ensuring consistency within semantic regions, which is tailored to the mask refinement task.

Our experiments demonstrate that Phoenix significantly outperforms existing refinement methods Lin et al. (2025); Tang et al. (2021); Wang et al. (2023); Yuan et al. (2020) across diverse tasks. When refining masks from semi-supervised and weakly-supervised models, Phoenix achieves absolute gains of up to +16.1% in $AP^{mask}$. Applied to state-of-the-art segmentation models, it consistently improves performance across diverse architectures. Furthermore, we demonstrate Phoenix's effectiveness in fine-grained mask refinement tasks and its potential for self-supervised refinement without ground-truth annotations.

## 2 RELATED WORK

**Image Segmentation** Image segmentation has evolved significantly with the advent of deep learning. Early approaches like FCN Long et al. (2015) and U-Net Ronneberger et al. (2015) established fully-convolutional architectures that became foundational for subsequent research. Performance improvements followed through innovations in feature extraction (*e.g.*, DeepLab Chen et al. (2018), PSPNet Zhao et al. (2017)) and, more recently, with transformer-based architectures (*e.g.*, Seg-Former Xie et al. (2021a), Mask2Former Cheng et al. (2022)) that leverage global context modeling. The recent Segment Anything Model (SAM) Kirillov et al. (2023) represents a significant advance in general-purpose segmentation through its prompt-based inference and zero-shot capabilities.

Despite these advances, generating high-quality segmentation masks remains challenging, particularly at object boundaries and in complex scenes. This challenge is magnified in the semi-supervised Wang et al. (2022) and weakly semi-supervised Kim et al. (2023) settings, where initial segmentation predictions are often coarse and contain substantial noise.

**Mask Refinement** Mask refinement addresses the limitations of primary segmentation models by enhancing mask quality through post-processing. Early approaches employed traditional techniques like conditional random fields (CRF) Krähenbühl & Koltun (2011) and graph cuts Boykov & Jolly (2001) to improve boundary adherence. These methods often rely on handcrafted features and struggle with semantically complex scenes. Learning-based refinement has gained traction with various approaches. SegFix Yuan et al. (2020) employs residual correction for boundaries, BPR Tang et al. (2021) introduces a boundary patch refinement that focuses on local regions around object boundaries. SegRefiner Wang et al. (2023) adopts a diffusion process for mask refinement modeling with a two-stage approach combining coarse prediction and fine-grained refinement. Recently, SAMRefiner Lin et al. (2025) leverages SAM's zero-shot capabilities through visual prompt engineering for mask refinement. Despite its effectiveness with a training-free setup, it relies solely on fixed pre-trained representations that weren't optimized for the refinement task. In contrast, Phoenix utilizes SAM's architecture with efficient fine-tuning techniques specifically designed to address key refinement challenges, such as correction pattern learning and boundary precision.

The primary challenge in effective mask refinement modeling lies in generating representative training data that captures realistic error patterns. Existing approaches Wang et al. (2023); Yuan et al. (2020) predominantly rely on morphological perturbations of ground-truth masks to simulate noise, such as random boundary perturbations with dilation and erosion operations and region modifications. These methods produce synthetic noise patterns that fail to capture the semantic nature of errors in real segmentation models, which are highly structured and context-dependent.

**Adversarial Learning** Adversarial learning has primarily focused on attack and defense mechanisms for model robustness Goodfellow et al. (2015); Madry et al. (2018). Adversarial attacks on segmentation models Arnab et al. (2018) and black-box perturbation methods Huang & Zhang (2020) demonstrate how adversarial techniques can expose model vulnerabilities through carefully crafted perturbations. However, existing work treats adversarial perturbations as destructive tools for testing rather than constructive mechanisms for training data generation. We introduce a paradigm shift by repurposing adversarial attack techniques as data augmentation for mask refinement, operating in the embedding space to generate semantically meaningful noise patterns that mimic realistic segmentation errors instead of merely maximizing prediction failures.

**Contrastive Learning** Contrastive learning has achieved remarkable success in representation learning Chen et al. (2020); He et al. (2020) with recent extensions to dense prediction tasks. Wang & Isola (2020) demonstrate that contrastive objectives in feature space can improve dense prediction quality through alignment and uniformity principles. Pixel-wise contrastive methods Wang et al. (2021); Xie et al. (2021b) have been developed for semantic segmentation pre-training and unsupervised representation learning. However, existing approaches primarily focus on learning general visual representations or establishing class boundaries. In contrast, our Contrastive Mask Refinement Learning introduces a novel tri-directional framework specifically designed for modeling the refinement relationship between noisy inputs, predictions, and ground truth masks, explicitly capturing the transformation from incorrect to correct predictions through self-improvement regularization.

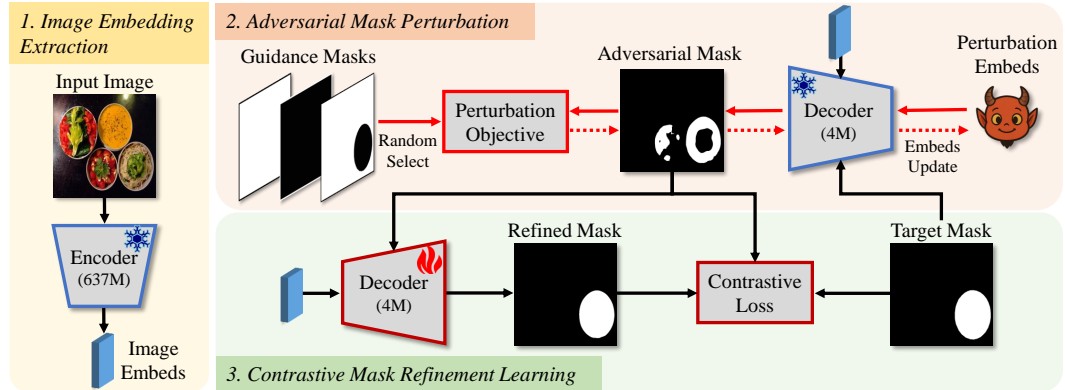

Figure 3: **Overview** of our Phoenix framework. The pipeline consists of three main components: (1) Image Embedding Extraction using SAM's encoder, (2) Adversarial Mask Perturbation that generates realistic noise patterns through adversarial embedding attacks, and (3) Contrastive Mask Refinement Learning that uses the tri-directional relationships between masks to improve refinement quality.

## 3 METHODOLOGY

### 3.1 PROBLEM FORMULATION

Mask refinement aims to transform a noisy or coarse segmentation mask into a high-quality mask that accurately delineates object boundaries and semantic regions. Formally, given an input image $\mathcal{I} \in \mathbb{R}^{H \times W \times 3}$ and a corresponding noisy mask $\mathcal{M}_n \in \{0,1\}^{H \times W}$, the mask refiner $f$ generates a refined mask $\mathcal{M}_r \in \{0,1\}^{H \times W}$ such that: $\mathcal{M}_r = f(\mathcal{I}, \mathcal{M}_n; \theta)$, where $\theta$ represents the parameters of the refiner model, and $H$ and $W$ denote the height and width of the image.

### 3.2 FRAMEWORK OVERVIEW

Figure 3 illustrates our Phoenix framework consisting of two key innovations: (1) Adversarial Mask Perturbation (AMP) and (2) Contrastive Mask Refinement Learning (CMRL). Phoenix builds upon pre-trained SAM Kirillov et al. (2023) with fine-tuning only the lightweight decoder. The encoder $f_{enc}$ takes an input image $\mathcal{I}$ and extracts image embeddings $\mathbf{E}_{img}$. The decoder $f_{dec}$ then processes these embeddings along with the noisy mask $\mathcal{M}_n$ to produce the refined mask $\mathcal{M}_r$. In addition, we extract point and box prompts from the noisy mask and incorporate them into the decoder's input as visual prompt embeddings $\mathbf{E}_v$, following the prompt sampling strategy used in Lin et al. (2025).

### 3.3 ADVERSARIAL MASK PERTURBATION (AMP)

**Limitations of Morphological Noise Approaches.** Existing mask refinement methods Wang et al. (2023); Yuan et al. (2020) predominantly rely on morphological perturbations (erosion, dilation, boundary modifications) to create synthetic noise from ground-truth masks. These approaches have several limitations, as evident in Figure 2a: (1) They produce unrealistic noise patterns that fail to capture the semantic nature of errors in real segmentation models, as morphological operations create structurally simplistic and spatially random noise, (2) They generate noise with limited diversity, unable to represent the wide range of failure modes in modern segmentation models, and (3) They are contextually blind, with perturbations operating independently of image content, making them unable to simulate errors from semantic confusion between similar objects.

**Proposed Adversarial Approach.** We introduce Adversarial Mask Perturbation (AMP), which leverages adversarial embedding attacks to generate semantically meaningful, contextually aware noise patterns. Given a target (ground-truth) mask $\mathcal{M}_t$, we inject learnable perturbation embeddings $\mathbf{E}_p \in \mathbb{R}^{P \times C}$ into the decoder $f_{dec}$ alongside visual prompt embeddings $\mathbf{E}_v$ derived from $\mathcal{M}_t$, where $P$ is the number of embeddings and $C$ is the embedding dimension. Importantly, the perturbation embeddings $\mathbf{E}_p$ are used exclusively for generating noisy masks during data augmentation without affecting the pretrained decoder parameters. The decoder $f_{dec}$ remains frozen during perturbation, and the $\mathbf{E}_p$ are not involved in actual training or inference of the decoder.

**Algorithm 1** Adversarial Mask Perturbation

**Require:** Image embeds $\mathbf{E}_{img}$, perturbation embeds $\mathbf{E}_p$, visual prompt embeds $\mathbf{E}_v$, target mask $\mathcal{M}_t$, guidance mask $\mathcal{M}_g$, adversarial criterion $\mathcal{D}$, initial step size $\alpha_0$, IoU threshold $\tau$, margin $\epsilon$, maximum inner iterations $N$.

**Ensure:** Noisy mask $\mathcal{M}_n$ with IoU $[\tau, \tau + \epsilon]$
1: $\alpha = \alpha_0$
2: **repeat**
3:   **for** $i = 1$ to $N$ **do**
4:     $\mathcal{M}_n = f_{dec}(\mathbf{E}_{img}, [\mathbf{E}_p; \mathbf{E}_v])$
5:     $iou = \text{compute\_IoU}(\mathcal{M}_n, \mathcal{M}_t)$
6:     **if** $iou < \tau + \epsilon$ **then**
7:       **break**
8:     **end if**
9:     Save current state: $\mathbf{E}_p^{prev} = \mathbf{E}_p$
10:     $\mathcal{L}_{adv} = -\mathcal{D}(\mathcal{M}_n, \mathcal{M}_g)$
11:     Compute gradient $\nabla_{\mathbf{E}_p} \mathcal{L}_{adv}$
12:     $\mathbf{E}_p = \mathbf{E}_p + \alpha \cdot \text{sign}(\nabla_{\mathbf{E}_p} \mathcal{L}_{adv})$
13:   **end for**
14:   **if** $iou \geq \tau$ **then**
15:     **return** $\mathcal{M}_n$ {Target IoU achieved}
16:   **end if**
17:   $\alpha = \alpha/10$ {Decay step size}
18:   Restore previous state: $\mathbf{E}_p = \mathbf{E}_p^{prev}$
19: **until** maximum decay steps reached
20: **return** $\mathcal{M}_n$

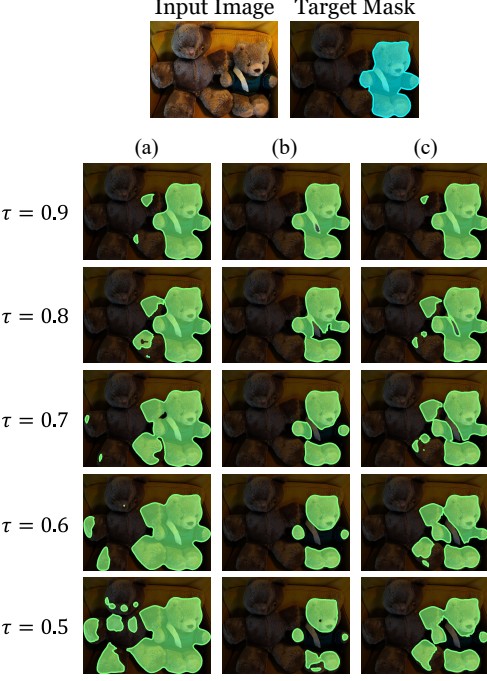

Figure 4: **Qualitative Samples** of generated noisy masks according to the IoU threshold $\tau$ and guidance mask (a) expansion guide, (b) contraction guide, and (c) inversion guide.

Unlike conventional adversarial attacks Arnab et al. (2018); Goodfellow et al. (2015); Madry et al. (2018) that aim to maximize classification error by perturbing input images, our approach repurposes adversarial techniques as constructive tools for noise generation. Specifically, we adapt the FGSM Goodfellow et al. (2015) to operate in the embedding space rather than image pixel space: $\mathbf{E}_p \leftarrow \mathbf{E}_p + \alpha \cdot \text{sign}(\nabla_{\mathbf{E}_p} \mathcal{L}_{adv})$, where $\alpha$ controls the perturbation magnitude and $\mathcal{L}_{adv}$ is an adversarial objective. By operating in the embedding space rather than input image space, we achieve both higher computational efficiency and high-level semantic perturbation Huang & Zhang (2020).

**Controllable Noise Generation.** The *Guidance Mask* ($\mathcal{M}_g$) determines the semantic direction of perturbation by modifying the adversarial objective: $\mathcal{L}_{adv} = -\mathcal{D}(f_{dec}(\mathbf{E}_{img}, [\mathbf{E}_p; \mathbf{E}_v]), \mathcal{M}_g)$, where $\mathcal{D}$ represents an adversarial criterion (*e.g.*, Dice or MSE loss). Three primary configurations yield distinct noise patterns, as shown in Figure 4: (1) *Expansion Guide* ($\mathcal{M}_g = \mathbf{1}$, all ones) generates false-positive errors by pushing boundaries outward, (2) *Contraction Guide* ($\mathcal{M}_g = \mathbf{0}$, all zeros) creates false-negative errors that mimic under-segmentation behaviors, (3) *Inversion Guide* ($\mathcal{M}_g = 1 - \mathcal{M}_t$) produces a balanced distribution of both error types.

Furthermore, to automatically calibrate the noise magnitude, Algorithm 1 implements an adaptive threshold-guided noise generation approach that adjusts the perturbation strength based on the IoU threshold parameter $\tau$. Given initial step size $\alpha_0$, IoU threshold $\tau$, margin $\epsilon$, and maximum inner iterations $N$, it iteratively updates $\mathbf{E}_p$ according to the FGSM rule, the process continues until the IoU between the generated noisy mask and the target mask falls within the range $[\tau, \tau + \epsilon]$. If this condition is not met until N iterations, it reduces the step size $\alpha \leftarrow \alpha/10$ and restarts the process. As shown in Figure 4, the IoU threshold $\tau$ enables control over noise intensity: lower thresholds generate more aggressive noise patterns, while higher thresholds produce subtle noise.

**Theoretical Analysis of Semantic Distribution.** For a pretrained decoder $f_{dec}$ with fixed parameters $\theta_{dec}$, the embedding gradient magnitude $\|\nabla_{\mathbf{E}_p} f_{dec}(\mathbf{E}_{img}, [\mathbf{E}_p; \mathbf{E}_v]; \theta_{dec})\|$ varies across spatial locations based on the model's uncertainty. Formally, this gradient magnitude relates to the local classification uncertainty: $\|\nabla_{\mathbf{E}_p} f_{dec}(\mathbf{E}_{img}, [\mathbf{E}_p; \mathbf{E}_v]; \theta_{dec})\|_2 \propto -\log p(y|\mathbf{E}_{img}, [\mathbf{E}_p; \mathbf{E}_v]; \theta_{dec})$, where $p(y|\mathbf{E}_{img}, [\mathbf{E}_p; \mathbf{E}_v]; \theta_{dec})$ represents the decoder's confidence in its prediction $y$ at each spatial location Kendall & Gal (2017). Consequently, our FGSM-based update intrinsically amplifies

perturbations in regions of high uncertainty, precisely where segmentation models typically struggle, producing semantically meaningful noise patterns.

We quantify the semantic nature of our generated noise through semantic correlation analysis between the perturbation magnitude and image feature gradients using the LVIS dataset Gupta et al. (2019). This analysis computes Pearson correlation Pearson (1895) between noise spatial distribution and semantic features extracted from edge detection and texture maps to measure how perturbations relate to image semantics (detailed in Appendix B.5). As shown in Figure 2c, adversarial noise exhibits a broad distribution of semantic correlations spanning $[-0.6, 0.8]$, indicating its ability to capture both semantically important regions (positive values) and homogeneous areas (negative values). In contrast, morphological noise (Figure 2b) shows a narrow distribution concentrated around zero, confirming its semantic-agnostic nature. This quantitative analysis validates that AMP produces noise patterns that align with the complex error distributions.

**Computational Efficiency.** Our implementation maintains high computational efficiency by reusing image embeddings from the encoder's single forward pass (637M parameters for ViT-H). Since AMP operates only on the lightweight decoder (4M parameters, 1.01 GFlops), each perturbation update requires only 6 ms on a V100 GPU, enabling efficient noise generation.

### 3.4 CONTRASTIVE MASK REFINEMENT LEARNING (CMRL)

**Motivation.** Traditional refinement approaches Tang et al. (2021); Wang et al. (2023); Yuan et al. (2020) primarily rely on pixel-wise classification, which treats each pixel independently. The fundamental challenge in mask refinement lies in modeling complex error patterns and their relationship to correct image contexts. Building on advances in contrastive learning, we propose a Contrastive Mask Refinement Learning (CMRL) approach that explicitly models the relationships between ground truth, noisy input, and model output masks. While conventional contrastive methods Chen et al. (2020); He et al. (2020) focus on representation learning by contrasting positive and negative pairs, our tri-directional approach establishes a more complex and unique relationship structure across three different masks and is designed to explicitly guide the mask refinement learning, which is tailored specifically for the mask refinement task.

**Formulation.** CMRL is designed with a tri-directional contrastive framework to address three objectives: (1) maintaining clear separation between foreground and background features, (2) ensuring consistency among features within the same semantic region, and (3) enabling self-improvement through bootstrapping from successful refinements. First, CMRL categorizes pixels into six distinct regions based on their classification in target ($\mathcal{M}_t$), noisy ($\mathcal{M}_n$), and refined ($\mathcal{M}_r$) masks:

$$\mathcal{T}_{fg} = (\mathcal{M}_t = 1) \wedge (\mathcal{M}_n = 1) \wedge (\mathcal{M}_r = 1), \quad \mathcal{T}_{bg} = (\mathcal{M}_t = 0) \wedge (\mathcal{M}_n = 0) \wedge (\mathcal{M}_r = 0)$$
$$\mathcal{S}_{fg} = (\mathcal{M}_t = 1) \wedge (\mathcal{M}_n = 0) \wedge (\mathcal{M}_r = 1), \quad \mathcal{S}_{bg} = (\mathcal{M}_t = 0) \wedge (\mathcal{M}_n = 1) \wedge (\mathcal{M}_r = 0)$$
$$\mathcal{F}_{fg} = (\mathcal{M}_t = 1) \wedge (\mathcal{M}_n = 0) \wedge (\mathcal{M}_r = 0), \quad \mathcal{F}_{bg} = (\mathcal{M}_t = 0) \wedge (\mathcal{M}_n = 1) \wedge (\mathcal{M}_r = 1)$$

where $\mathcal{T}$, $\mathcal{S}$, and $\mathcal{F}$ denote true, success, and failure regions, with $fg$ and $bg$ indicating foreground and background. This categorization creates a natural curriculum where the model progressively refines its predictions by learning from both initially correct regions and its own successful refinements.

**Notation.** Let $\mathbf{F}$ denote upsampled image embeddings. We apply projector $g$ consisting of a 3-layer MLP to obtain projection feature maps $\mathbf{p} = g(\mathbf{F}) \in \mathbb{R}^{c \times h \times w}$. For each position $i \in \Omega = \{1, \ldots, h \times w\}$, $\mathbf{p}_i \in \mathbb{R}^c$ denotes the feature vector at position $i$. The expectation $\mathbb{E}_{i \in \mathcal{R}}[\cdot]$ denotes uniform sampling over pixels in region $\mathcal{R}$. The similarity function $\text{sim}(\mathbf{p}_i, \mathbf{p}_j) = \mathbf{p}_i^\top \mathbf{p}_j$ computes cosine similarity between L2-normalized features. Unlike conventional segmentation losses that operate in the pixel space, our contrastive framework operates in the feature space, enabling the model to learn rich feature representations that capture contextual information Wang & Isola (2020).

**Intra-Class Feature Consistency** aims to create coherent feature representations within each semantic class, ensuring that all parts of the same object share similar feature characteristics regardless of their visual appearance. For foreground features, the intra-class loss is defined as:

$$\mathcal{L}_{intra}^{fg} = -\mathbb{E}_{i \in \mathcal{F}_{fg}} \left[ \log \frac{\sum_{j \in \mathcal{S}_{fg} \cup \mathcal{T}_{fg}} \exp(\text{sim}(\mathbf{p}_i, \mathbf{p}_j)/\tau)}{\sum_{k \in \Omega} \exp(\text{sim}(\mathbf{p}_i, \mathbf{p}_k)/\tau)} \right] \tag{1}$$

where $\tau$ is a temperature parameter. Following the InfoNCE contrastive loss Oord et al. (2018), this loss maximizes the similarity between foreground failure features and correct foreground features

Table 1: Performance of refined masks on COCO train5K using LVIS annotations. We denote full mask annotations as $\mathcal{F}$, unlabeled data as $\mathcal{U}$, and (object center) point annotations as $\mathcal{P}$.

(a) Semi-Supervised, NB Wang et al. (2022)

| Methods | Annotations | AP$^{mask}$ | AP$^{boundary}$ |
|---|---|---|---|
| NB | | 5.1 | 1.9 |
| +SegRefiner | $\mathcal{F}$ 1% + $\mathcal{U}$ 99% | 6.2 | 3.8 |
| +SAMRefiner | | 8.1 | 6.0 |
| +Phoenix (ours) | | **9.8** (+4.7) | **8.1** (+6.2) |
| NB | | 18.5 | 9.7 |
| +SegRefiner | $\mathcal{F}$ 5% + $\mathcal{U}$ 95% | 20.4 | 14.0 |
| +SAMRefiner | | 23.8 | 18.5 |
| +Phoenix (ours) | | **26.7** (+8.2) | **22.2** (+12.5) |
| NB | | 22.6 | 12.7 |
| +SegRefine | $\mathcal{F}$ 10% + $\mathcal{U}$ 90% | 25.0 | 17.9 |
| +SAMRefiner | | 28.2 | 22.1 |
| +Phoenix (ours) | | **30.8** (+8.2) | **25.7** (+13.0) |

(b) Weakly Semi-Supervised, PointWSSIS Kim et al. (2023)

| Methods | Annotations | AP$^{mask}$ | AP$^{boundary}$ |
|---|---|---|---|
| PointWSSIS | | 12.6 | 6.3 |
| +SegRefiner | $\mathcal{F}$ 1% + $\mathcal{P}$ 99% | 14.7 | 9.5 |
| +SAMRefiner | | 21.8 | 16.4 |
| +Phoenix (ours) | | **28.7** (+16.1) | **23.6** (+17.3) |
| PointWSSIS | | 25.7 | 16.4 |
| +SegRefiner | $\mathcal{F}$ 5% + $\mathcal{P}$ 95% | 27.6 | 20.2 |
| +SAMRefiner | | 32.8 | 26.2 |
| +Phoenix (ours) | | **36.3** (+10.6) | **30.2** (+13.8) |
| PointWSSIS | | 30.2 | 20.4 |
| +SegRefiner | $\mathcal{F}$ 10% + $\mathcal{P}$ 90% | 31.7 | 23.8 |
| +SAMRefiner | | 36.6 | 29.7 |
| +Phoenix (ours) | | **38.9** (+8.7) | **32.6** (+12.2) |

Table 2: Performance of refined masks on COCO validation set using LVIS annotations.

(a) Results on Mask R-CNN

| Method | AP$^{mask}$ | AP$^{boundary}$ |
|---|---|---|
| MRCNN(RN50) | 39.8 | 27.3 |
| +SegFix | 40.6 | 29.1 |
| +BPR | 41.0 | 30.4 |
| +SegRefiner | 41.9 | 32.6 |
| +SAMRefiner | 45.3 | 35.9 |
| +Phoenix (ours) | **46.9** (+7.1) | **38.8** (+11.5) |
| MRCNN(RN101) | 41.6 | 29.0 |
| +SegFix | 42.2 | 30.6 |
| +BPR | 42.8 | 32.0 |
| +SegRefiner | 43.6 | 34.1 |
| +SAMRefiner | 46.6 | 36.9 |
| +Phoenix (ours) | **48.1** (+6.5) | **39.8** (+10.8) |

(b) Results on State-of-the-art Segmentation Models

| Method | AP$^{mask}$ | AP$^{boundary}$ | Method | AP$^{mask}$ | AP$^{boundary}$ |
|---|---|---|---|---|---|
| SOLO | 37.4 | 24.7 | CondInst | 39.8 | 29.2 |
| +SegRefiner | 40.5 | 31.3 | +SegRefiner | 41.1 | 32.2 |
| +SAMRefiner | 44.1 | 34.2 | +SAMRefiner | 45.2 | 35.8 |
| +Phoenix (ours) | **46.2** (+8.8) | **37.7** (+13.0) | +Phoenix (ours) | **46.7** (+6.9) | **38.6** (+9.4) |
| RefineMask | 41.2 | 30.5 | Mask2Former | 46.8 | 37.0 |
| +SegRefiner | 41.9 | 33.0 | +SegRefiner | 47.4 | 38.8 |
| +SAMRefiner | 44.7 | 35.3 | +SAMRefiner | 49.0 | 39.0 |
| +Phoenix (ours) | **46.0** (+4.8) | **37.9** (+7.4) | +Phoenix (ours) | **50.6** (+3.8) | **42.1** (+5.1) |
| ViTDet | 54.6 | 42.5 | MaskDINO | 56.8 | 46.5 |
| +SegRefiner | 55.5 | 46.0 | +SegRefiner | 57.0 | 47.7 |
| +SAMRefiner | 55.8 | 46.2 | +SAMRefiner | 57.0 | 47.4 |
| +Phoenix (ours) | **56.3** (+1.7) | **46.7** (+4.2) | +Phoenix (ours) | **58.0** (+1.2) | **48.6** (+2.1) |

while implicitly pushing away features from other regions. A similar loss $\mathcal{L}_{intra}^{bg}$ is computed for background features, and the total intra-class loss is $\mathcal{L}_{intra} = \mathcal{L}_{intra}^{fg} + \mathcal{L}_{intra}^{bg}$.

**Inter-Class Feature Contrast** enforces clear separation between foreground and background features, particularly in error-prone regions, by maximizing the distance between the two features:

$$\mathcal{L}_{inter}^{fg \to bg} = \mathbb{E}_{i \in \mathcal{F}_{fg}} \left[ \log \sum_{j \in \mathcal{F}_{bg} \cup \mathcal{S}_{bg} \cup \mathcal{T}_{bg}} \exp(\text{sim}(\mathbf{p}_i, \mathbf{p}_j)/\tau) \right] \quad (2)$$

It pushes foreground failure features away from all background features. Likewise, $\mathcal{L}_{inter}^{bg \to fg}$ is computed. By maximizing the feature distance between different classes, this bidirectional repulsion reshapes the feature space to create clearer decision boundaries between foreground and background. The total inter-class loss is $\mathcal{L}_{inter} = \mathcal{L}_{inter}^{fg \to bg} + \mathcal{L}_{inter}^{bg \to fg}$.

**Self-Improvement Regularization** leverages successfully refined regions to guide the improvement of currently unrefined errors, creating a learning pathway from failure to success within the same image. This component enables the model to learn from its own successful corrections, addressing the unique challenge of transforming incorrect predictions into correct ones.

$$\mathcal{L}_{self} = -\mathbb{E}_{i \in \mathcal{F}_{fg} \cup \mathcal{F}_{bg}} \left[ \log \frac{\sum_{j \in \mathcal{S}_{fg} \cup \mathcal{S}_{bg}} \exp(\text{sim}(\mathbf{p}_i, \mathbf{p}_j)/\tau)}{\sum_{k \in \Omega} \exp(\text{sim}(\mathbf{p}_i, \mathbf{p}_k)/\tau)} \right] \quad (3)$$

This loss encourages features from current failure regions ($\mathcal{F}_{fg} \cup \mathcal{F}_{bg}$) to resemble those from successfully refined regions ($\mathcal{S}_{fg} \cup \mathcal{S}_{bg}$) within the same image. Unlike the other components that focus on static correctness, this loss explicitly models the transformation from incorrect to correct predictions, creating a bootstrapping mechanism where the model learns from its own successful refinements to improve regions that are still incorrectly classified.

**Loss Function.** Our final CMRL loss combines all three objectives with weighting parameters:

$$\mathcal{L}_{CMRL} = \lambda_{intra} \cdot \mathcal{L}_{intra} + \lambda_{inter} \cdot \mathcal{L}_{inter} + \lambda_{self} \cdot \mathcal{L}_{self} \quad (4)$$

where $\lambda_{intra}$, $\lambda_{inter}$, and $\lambda_{self}$ balance the respective objectives. This contrastive loss is combined with a traditional segmentation loss to produce the final training objective.

Table 3: Performance of refined masks on the DIS task using coarse masks from 4 different models. SAMRefiner[†] means using the HQ-SAM Ke et al. (2023) backbone for finer mask prediction.

| Methods | DIS-VD | | DIS-TE1 | | DIS-TE2 | | DIS-TE3 | | DIS-TE4 | | *Average* | |
|---|---|---|---|---|---|---|---|---|---|---|---|---|
| | IoU | Boundary $\mathcal{F}$ | IoU | Boundary $\mathcal{F}$ | IoU | Boundary $\mathcal{F}$ | IoU | Boundary $\mathcal{F}$ | IoU | Boundary $\mathcal{F}$ | IoU | Boundary $\mathcal{F}$ |
| U-Net | 54.8 | 67.0 | 44.1 | 59.2 | 54.4 | 66.1 | 59.3 | 72.0 | 61.1 | 77.8 | 54.7 | 68.4 |
| +SAMRefiner[†] | 63.7 | 72.4 | 56.5 | 74.3 | 67.1 | 76.3 | 69.3 | 74.4 | 64.8 | 64.8 | 64.3 | 72.4 |
| +SegRefiner | 58.7 | 71.0 | 47.0 | 63.4 | 58.1 | 70.7 | 63.4 | 75.9 | 66.3 | 81.8 | 58.7 | 72.6 |
| +Phoenix (ours) | 74.9 (+20.1) | 84.2 (+17.2) | 69.6 (+25.5) | 83.3 (+24.1) | 79.3 (+24.9) | 85.8 (+19.7) | 79.4 (+20.1) | 85.8 (+13.8) | 75.2 (+14.1) | 82.9 (+5.1) | 75.7 (+21.0) | 84.4 (+16.0) |
| PSPNet | 56.4 | 69.3 | 47.6 | 66.8 | 57.7 | 70.5 | 59.9 | 71.8 | 57.6 | 70.4 | 55.8 | 69.8 |
| +SAMRefiner[†] | 64.2 | 71.7 | 58.8 | 76.4 | 67.2 | 76.6 | 68.0 | 72.5 | 63.2 | 62.9 | 64.3 | 72.0 |
| +SegRefiner | 61.9 | 73.7 | 50.4 | 69.0 | 62.0 | 74.3 | 65.3 | 77.5 | 66.7 | 80.3 | 61.3 | 75.0 |
| +Phoenix (ours) | 75.7 (+19.3) | 84.8 (+15.5) | 70.3 (+22.7) | 84.0 (+17.2) | 79.5 (+21.8) | 86.0 (+15.5) | 79.8 (+19.9) | 86.2 (+14.4) | 74.4 (+16.8) | 82.8 (+12.4) | 75.9 (+20.1) | 84.8 (+15.0) |
| HRNet | 61.0 | 74.2 | 51.0 | 67.1 | 61.2 | 73.6 | 64.4 | 77.9 | 64.7 | 82.0 | 60.5 | 74.9 |
| +SAMRefiner[†] | 67.5 | 74.2 | 59.7 | 76.6 | 68.1 | 77.9 | 72.6 | 76.0 | 65.8 | 65.0 | 66.8 | 73.9 |
| +SegRefiner | 64.4 | 76.1 | 52.9 | 68.7 | 64.7 | 75.7 | 67.6 | 79.8 | 70.5 | 84.6 | 64.0 | 77.0 |
| +Phoenix (ours) | 76.5 (+15.5) | 85.8 (+11.6) | 69.6 (+18.6) | 83.4 (+16.3) | 78.1 (+16.9) | 86.4 (+12.8) | 80.7 (+16.3) | 86.6 (+8.7) | 74.9 (+10.2) | 83.4 (+1.4) | 76.0 (+15.5) | 85.1 (+10.2) |
| ISNet | 67.1 | 79.8 | 56.2 | 74.9 | 67.1 | 78.4 | 69.9 | 81.2 | 70.1 | 79.6 | 66.1 | 79.6 |
| +SAMRefiner[†] | 68.1 | 75.2 | 60.6 | 78.4 | 69.4 | 78.2 | 71.7 | 74.8 | 66.0 | 64.0 | 67.1 | 74.1 |
| +SegRefiner | 68.3 | 80.2 | 56.9 | 75.0 | 67.8 | 79.0 | 70.9 | 81.9 | 72.2 | 85.2 | 67.2 | 80.3 |
| +Phoenix (ours) | 76.7 (+9.6) | 86.1 (+6.3) | 71.4 (+17.7) | 85.3 (+10.4) | 79.3 (+12.2) | 86.7 (+8.3) | 80.2 (+10.3) | 87.0 (+5.8) | 75.3 (+5.2) | 84.2 (+0.4) | 76.6 (+11.0) | 85.9 (+6.3) |

## 4 EXPERIMENTS

### 4.1 EXPERIMENTAL SETUP

**Datasets.** For general object mask refinement, we train Phoenix on the LVIS Gupta et al. (2019) dataset following the same setup as SegRefiner Wang et al. (2023). We evaluate on masks from both low-quality and high-quality segmentation models, refining coarse masks generated by semi-supervised (NB Wang et al. (2022)) and weakly semi-supervised (PointWSSIS Kim et al. (2023)) methods by measuring the quality of pseudo labels on the COCO Train5K dataset Kim et al. (2023). We also evaluate on masks produced by various instance segmentation models on the COCO Lin et al. (2014) validation set. Separately, for the fine-grained segmentation task, we train Phoenix on DIS5K Qin et al. (2022) and ThinObject-5K Liew et al. (2021) datasets and evaluate it on the DIS task Qin et al. (2022), which demands precise boundary delineation for thin structures, complex topologies, and fine-grained details.

**Evaluation Metrics.** We use Average Precision (AP) and boundary AP ($AP^{boundary}$) Cheng et al. (2021) to evaluate instance segmentation quality, and Intersection over Union (IoU) and Boundary $\mathcal{F}$ measure Perazzi et al. (2016) for fine-grained segmentation tasks.

**Implementation Details.** Our mask refiner builds upon the pre-trained SAM Kirillov et al. (2023) with the ViT-H Dosovitskiy et al. (2021) backbone. During training, we freeze the encoder (637M parameters) and fine-tune only the decoder (4M parameters). We optimize the model using AdamW Loshchilov & Hutter (2017) with an initial learning rate of $1 \times 10^{-4}$ and cosine decay scheduling. Training proceeds for 10K iterations with a total batch size of 16 on 8 V100 GPUs, requiring less than 10 hours of total training time. For the adversarial mask perturbation process, we use Dice loss as the perturbation objective with the maximum inner iterations $N$ of 10, the initial step size $\alpha_0$ of 0.01, and the margin $\epsilon$ of 5%. The IoU threshold $\tau$ is randomly sampled from a uniform distribution between 0.3 and 0.9, and the guidance mask $\mathcal{M}_g$ is randomly chosen among expansion, contraction, and inversion guides. For contrastive loss weightings, we set $\lambda_{intra} = 0.4$, $\lambda_{inter} = 0.4$, and $\lambda_{self} = 0.2$ by default. The contrastive loss $\mathcal{L}_{CMRL}$ is added to the loss function used in SAM, which is the combination of Dice and Focal Lin et al. (2017) losses.

### 4.2 RESULTS ON INSTANCE SEGMENTATION

Table 1 presents the performance of various refinement methods on coarse masks generated by semi-supervised Wang et al. (2022) and weakly semi-supervised Kim et al. (2023) approaches. Our method consistently outperforms previous state-of-the-art refiners across all settings, largely due to our semantic-aware noise modeling and tri-directional contrastive learning. Phoenix achieves significant improvements when refining extremely noisy masks produced with minimal supervision (1% fully-annotated and 99% point labels), with absolute gains of up to +16.1% in $AP^{mask}$ and +17.3% in $AP^{boundary}$. Moreover, Table 2 shows our effectiveness in refining masks from various modern instance segmentation models He et al. (2017); Kirillov et al. (2020); Tian et al. (2020); Cheng et al. (2022); Li et al. (2022; 2023a). Phoenix achieves superior performance to existing refiners in all settings, demonstrating the versatility of our refinement approach across model architectures.

Table 4: **Ablation Study** using instance segmentation results on 1% PointWSSIS ($AP^1$) and MRCNN with a ResNet-50 backbone ($AP^2$) and fine-grained segmentation results on DIS-UNet ($IoU^1$) and DIS-ISNet ($IoU^2$), averaged across all DIS tasks. We denote morphological perturbation as *morp* and adversarial perturbation as *adv*.

(a) $\tau$ (IoU Threshold)

| $\tau$ | $AP^1$ | $AP^2$ |
|---|---|---|
| 0.3 | 28.2 | 45.5 |
| 0.5 | 27.3 | 46.4 |
| 0.7 | 25.3 | 46.6 |
| 0.9 | 24.2 | 46.2 |
| $\mathcal{U}(0.3, 0.9)$ | **28.7** | **46.9** |

(b) $\mathcal{D}$ (Adversarial Criterion)

| Adv Func | $AP^1$ | $AP^2$ |
|---|---|---|
| L1 | 27.7 | 46.5 |
| MSE | 28.7 | 46.8 |
| BCE | 28.4 | 46.8 |
| Focal | 28.2 | 46.6 |
| Dice | **28.7** | **46.9** |

(c) Contrastive Loss Component

| $\mathcal{L}_{intra}$ | $\mathcal{L}_{inter}$ | $\mathcal{L}_{self}$ | $AP^1$ | $AP^2$ |
|---|---|---|---|---|
| ✗ | ✗ | ✗ | 26.9 | 46.1 |
| ✓ | ✗ | ✗ | 28.1 | 46.6 |
| ✗ | ✓ | ✗ | 28.2 | 46.6 |
| ✓ | ✓ | ✗ | 28.5 | 46.8 |
| ✓ | ✓ | ✓ | **28.7** | **46.9** |

(d) Mask Perturbation

| Method | Perturb | $IoU^1$ | $IoU^2$ |
|---|---|---|---|
| SegRefiner | Morp | 58.7 | 74.2 |
| SegRefiner | Adv | 71.8 | 76.6 |
| Phoenix | Morp | 67.8 | 71.0 |
| Phoenix | Adv | **75.5** | **77.1** |

(e) Noise from Model

| Noisy Mask | $IoU^1$ | $IoU^2$ |
|---|---|---|
| UNet | 67.6 | 71.6 |
| ISNet | 65.6 | 70.2 |
| Adv (ours) | **75.7** | **79.3** |

(f) Component Analysis

| Perturb | CMRL | $AP^1$ | $AP^2$ | $IoU^1$ | $IoU^2$ |
|---|---|---|---|---|---|
| Morp | ✗ | 22.3 | 45.2 | 65.3 | 69.9 |
| Adv | ✗ | 26.9 | 46.1 | 71.5 | 74.2 |
| Morp | ✓ | 23.8 | 45.5 | 67.8 | 71.0 |
| Adv | ✓ | **28.7** | **46.9** | **75.7** | **77.1** |

## 4.3 RESULTS ON FINE-GRAINED SEGMENTATION

Table 3 presents the performance of Phoenix when applied to the challenging DIS Qin et al. (2022) task across multiple test datasets and backbone models Ronneberger et al. (2015); Shen et al. (2022); Wang et al. (2020); Zhao et al. (2017). The results demonstrate that our method consistently outperforms both SAMRefiner and SegRefiner by substantial margins, with average improvements ranging from 11% to 21% in IoU, highlighting of the effectiveness of our realistic and contextual noise perturbation modeling in the mask refinement task. Figure 1 provides visual comparisons between our method and existing refiners. Phoenix produces masks with significantly improved boundary adherence and structural integrity compared to both noisy inputs and competing refiners, successfully recovering challenging scenarios, such as thin structures and intricate boundaries.

## 4.4 ABLATION STUDIES

We conduct extensive ablation studies to analyze the contribution of individual components and hyperparameters. For these experiments, we use low-quality masks from PointWSSIS with 1% supervision (denoted as $AP^1$) and high-quality masks from Mask R-CNN with ResNet-50 He et al. (2017) backbone (denoted as $AP^2$). Additionally, we evaluate fine-grained segmentation performance on DIS-UNet (denoted as $IoU^1$) and DIS-ISNet (denoted as $IoU^2$), averaged across all the DIS test splits. **Additional ablation studies and in-depth analyses can be found in the appendix.**

**Effect of Adversarial Mask Perturbation Parameters.** Table 4a shows the impact of the IoU threshold $\tau$ on refinement performance. Higher $\tau$ values lead to better performance on high-quality masks (higher $AP^2$) but lower performance on low-quality masks (lower $AP^1$). For robustness against various noise patterns, we randomly sample $\tau$ from a uniform distribution $\mathcal{U}(0.3, 0.9)$ for each noise mask generation step during training, achieving the best balanced performance. Moreover, Table 4b presents the performance with different adversarial criteria, demonstrating our method is relatively robust to the choice of objective, with Dice loss providing the best overall balance.

**Effect of Contrastive Loss Components.** Table 4c illustrates the performance impact of different contrastive loss components. We observe that both intra-class consistency ($\mathcal{L}_{intra}$) and inter-class contrast ($\mathcal{L}_{inter}$) contribute significantly to performance, with their combination yielding substantial improvements. Including all three components achieves the optimal performance.

**Effect of Mask Perturbation Method.** Table 4d compares different mask perturbation approaches across models. When applying adversarial perturbation to SegRefiner, its performance improves substantially. Conversely, when our Phoenix is trained with simple morphological perturbations, performance drops significantly compared to our full approach. These results highlight the critical importance of sophisticated noise generation in mask refinement learning.

**Comparison with Real Model Errors.** Table 4e compares our adversarially generated noise patterns with noise obtained directly from real segmentation models. While using output masks from the pre-trained model (UNet Ronneberger et al. (2015) or ISNet Shen et al. (2022)) as noisy masks for training provides adequate performance, our adversarial approach significantly outperforms them. This superiority stems from two key advantages: (1) a wider diversity of error patterns, and (2) controllable perturbation magnitude.

**Effect of Individual Components.** Table 4f isolates the contributions of AMP and CMRL. Starting from the baseline (morphological noise with pixel-wise loss), AMP alone provides substantial improvements (+4.6% AP[1], +6.2% IoU[1]), demonstrating the importance of semantic-aware noise generation. CMRL alone yields meaningful gains (+1.5% AP[1], +2.5% IoU[1]), validating the impact of our tri-directional contrastive framework. Combining both components achieves the best performance (+6.4% AP[1], +10.4% IoU[1]), with the combined improvement exceeding the sum of individual contributions (+8.7% IoU[1]), demonstrating true synergy. This synergistic effect occurs because CMRL is more effective when encountering realistic noise patterns from AMP; the semantically meaningful error distributions enable more informative tri-directional contrastive relationships, leading to superior refinement learning. Notably, AMP provides larger marginal gains on challenging scenarios, while both components contribute substantially across all settings, confirming their complementary nature.

## 5 CONCLUSION

We presented Phoenix, a novel framework for mask refinement that leverages adversarial perturbation and contrastive learning to address the limitations of existing approaches. Our method generates semantically meaningful noise patterns that better reflect real-world segmentation errors and employs a tri-directional contrastive learning approach that explicitly models the relationships between ground truth, noisy input, and current prediction. Extensive experiments demonstrate that Phoenix significantly outperforms existing methods across diverse tasks and noise conditions. We believe our work establishes a new foundation for mask refinement and opens up promising directions for future research, including self-supervised refinement and multimodal guidance integration.

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

APPENDIX

# A ADDITIONAL APPLICATIONS OF PHOENIX

## A.1 SELF-SUPERVISED MASK REFINEMENT LEARNING.

Our intriguing finding is the potential for self-supervised mask refinement without ground-truth masks. By leveraging SAM's zero-shot capability, we can generate pseudo-target masks using a grid of prompting points, enabling training with target and noisy mask pairs without ground-truth annotations. As shown in Table 5a, our self-supervised pipeline achieves competitive performance in $AP^1$ compared to the fully supervised setting. This success can be attributed to our contrastive learning framework, which focuses on the relationship between mask pairs and learns generalizable noise patterns rather than overfitting to specific annotations. However, this self-supervised approach has limitations. Its effectiveness depends on SAM's zero-shot capability for the target task. For fine-grained segmentation tasks where SAM struggles to generate high-quality masks, the self-supervised pipeline underperforms. Nevertheless, these results point to promising future directions in reducing annotation requirements for mask refinement.

## A.2 INTEGRATION OF VISUAL CUES.

We identify limitations when handling extremely challenging noise patterns, such as complete mislocalization and uncertain target objects. In these scenarios, refinement becomes inherently ambiguous without additional guidance. To address this, we explored integrating (ground-truth) visual prompts (box and points derived from the ground-truth mask) that provide explicit target object information, achieving remarkable performance improvements, as shown in Table 5b. Another important challenge is handling misclassified object masks. Since our approach operates in a class-agnostic manner, it cannot correct semantic class errors inherent from real models. To address this limitation, incorporating open vocabulary models Ghiasi et al. (2022); Liang et al. (2023) or text embeddings Li et al. (2023b); Radford et al. (2021) will be a promising research direction. These insights highlight both the current capabilities and future potential of mask refinement systems, pointing toward more robust, multimodal approaches that can handle increasingly diverse and challenging segmentation scenarios.

Table 5: **Ablation Study** using instance segmentation results on 1% PointWSSIS ($AP^1$) and fine-grained segmentation results on DIS-UNet ($IoU^1$) and DIS-ISNet ($IoU^2$), averaged across all DIS tasks. We denote the self-supervised mask refinement as *SSL*.

(a) Self-Supervised Mask Refinement Learning

| SSL | $AP^1$ | $IoU^1$ |
|---|---|---|
| ✗ | 28.7 | 75.7 |
| ✓ | 27.4 | 58.0 |

(b) Integration of Ground-Truth Visual Prompt

| Prompt | $AP^1$ | $IoU^1$ |
|---|---|---|
| - | 28.7 | 75.7 |
| Point | 30.3 | 77.0 |
| Box | 37.3 | 78.5 |
| Point+Box | 37.4 | 78.6 |

## A.3 SEMANTIC SEGMENTATION

We evaluate Phoenix on the PASCAL VOC2012 Everingham et al. (2010) semantic segmentation benchmark to demonstrate its effectiveness in the semantic segmentation domain. While our main paper focuses on instance and fine-grained segmentation, this evaluation aims to verify the versatility of our refinement approach across diverse segmentation tasks. For this evaluation, we follow the *split-then-merge* strategy used in SAMRefiner Lin et al. (2025) for mask refinement in the semantic segmentation domain and leverage the Phoenix model trained on the LVIS Gupta et al. (2019) instance segmentation dataset. This approach allows us to refine pseudo semantic labels generated by unsupervised Zhou et al. (2022) or weakly-supervised Lin et al. (2023); Rong et al. (2023) models on the VOC2012 training set, addressing the challenge of noisy pseudo-labels that often plague these learning paradigms. As shown in Table 6a, Phoenix consistently improves the performance of various semantic segmentation methods. Namely, Phoenix improves the unsupervised method, MaskCLIP Zhou et al. (2022), from 47.8% to 59.1% (+11.3%) in mIoU, effectively refining coarse

Table 6: **Evaluation of Phoenix on additional applications**. (a) Semantic segmentation results on PASCAL VOC2012 training set across different models: $\mathcal{U}$ (unlabeled) and $\mathcal{I}$ (image-level labels) supervisions. (b) Ground truth annotation refinement on COCO2017 validation set. The † denotes that the model is trained using the VOC2012 dataset.

(a) Results on VOC2012.

| Method | Annotations | mIoU |
|---|---|---|
| MaskCLIP | | 47.8 |
| +SAMRefiner | $\mathcal{U}$ | 57.3 |
| +Phoenix (ours) | | 59.1 (+11.3) |
| BECO | | 66.3 |
| +SAMRefiner | $\mathcal{I}$ | 71.8 |
| +Phoenix (ours) | | 75.0 (+8.7) |
| CLIP-ES | | 70.8 |
| +SAMRefiner | $\mathcal{I}$ | 79.3 |
| +Phoenix (ours) | | 81.6 (+10.8) |

(b) Refined masks on COCO2017 *val*.

| Data | $AP^{mask}$ | $AP^{boundary}$ |
|---|---|---|
| COCO val | 38.3 | 27.3 |
| +SAMRefiner | 41.5 | 33.0 |
| +Phoenix† (ours) | 42.2 (+3.9) | 33.6 (+6.3) |
| +Phoenix (ours) | 43.7 (+5.4) | 36.1 (+8.8) |

pseudo-labels to achieve more accurate semantic boundaries. In addition, Phoenix enhances the weakly-supervised methods, BECO Rong et al. (2023) and CLIP-ES Lin et al. (2023), by +8.7% and +10.8%, respectively. The qualitative results in Figure 6 demonstrate Phoenix's ability to correct both over-segmentation and under-segmentation errors in pseudo-labels, particularly at object boundaries and in regions with complex semantic transitions. These substantial gains highlight the universal applicability of our approach.

## A.4 HUMAN ANNOTATION CORRECTION

We investigate the effectiveness of Phoenix for improving manually annotated masks, which often contain imperfections, particularly at complex boundaries. This application is important because human annotations in datasets like COCO Lin et al. (2014) are frequently coarse due to the sparse polygon points format used during annotation, limiting boundary precision. We use the COCO dataset's annotations as representative of human-annotated masks and evaluate their quality against the more precise LVIS Gupta et al. (2019) annotations as reference. The LVIS dataset provides a finer version of COCO mask annotations through more detailed annotation protocols, which can be regarded as baseline refined masks. As shown in Table 6b, Phoenix significantly improves the annotation quality of COCO2017 *val* ground-truth masks, enhancing $AP^{mask}$ by +5.4% and $AP^{boundary}$ by +8.8%. The qualitative examples in Figure 7 illustrate how Phoenix refines coarse COCO annotations to better align with precise ground truth boundaries. The improvement is particularly pronounced at boundaries, where human annotators often struggle with precision. Our method effectively identifies and corrects these imperfections, suggesting potential applications in annotation workflow enhancement and dataset quality improvement. This capability could significantly reduce the time and cost associated with creating high-quality segmentation datasets by allowing annotators to focus on object identification rather than precise boundary delineation.

## A.5 ZERO-SHOT GENERALIZATION

To assess the generalization capabilities of Phoenix beyond its training domain, we evaluate our method in a zero-shot setting on two challenging datasets: Cityscapes Cordts et al. (2016) for urban scene understanding and the ISIC2018 skin lesion segmentation dataset Codella et al. (2019) for medical imaging. Importantly, Phoenix is trained exclusively on the LVIS dataset and applied directly to these domains without any domain-specific fine-tuning or adaptation.

**Cityscapes Evaluation** Table 7a shows the performance of Phoenix when applied to refine instance segmentation masks on the Cityscapes validation set. For the evaluation metric, we follow the COCO-style evaluation metrics Lin et al. (2014). Despite the substantial domain gap between LVIS and Cityscapes, Phoenix demonstrates robust generalization. Our method achieves consistent improvements over the baseline Mask R-CNN (R50) He et al. (2017), with gains of +1.1% in AP and +1.0% in AP50. Notably, while SAMRefiner shows degraded performance (-3.6% AP), Phoenix maintains and enhances the original segmentation quality, highlighting the effectiveness of our noise modeling approach across different visual domains.

**Medical Imaging Evaluation**    The results on the ISIC2018 skin lesion segmentation dataset (Table 7b) further demonstrate Phoenix's domain adaptability. Medical imaging presents unique challenges including varying illumination conditions, texture variations, and irregular lesion boundaries that differ substantially from natural images in LVIS. Despite these challenges, Phoenix achieves substantial improvements over the baseline UNet Ronneberger et al. (2015), with gains of +4.6% in IoU and +6.0% in F1 score. The superior performance compared to SAMRefiner (+3.7% IoU improvement) underscores the robustness of our tri-directional contrastive learning and semantic-aware noise modeling in handling domain-specific segmentation challenges. These zero-shot results validate that Phoenix learns generalizable refinement principles rather than dataset-specific patterns, making it a versatile solution for mask refinement across diverse application domains without requiring additional training or adaptation.

Table 7: **Zero-Shot Generalization of Phoenix on** (a) Cityscapes instance segmentation validation set and (b) Medical imaging, ISIC2018 skin lesion segmentation dataset.

(a) Results on Cityscapes

| Method | AP | AP50 |
|---|---|---|
| MRCNN (R50) | 36.1 | 60.9 |
| +SAMRefiner | 32.5 | 58.4 |
| +Phoenix (ours) | 37.2 (+1.1) | 61.9 (+1.0) |

(b) Results on Medical Imaging Dataset

| Method | IoU | F1 |
|---|---|---|
| UNet | 54.6 | 71.2 |
| +SAMRefiner | 55.5 | 76.1 |
| +Phoenix (ours) | 59.2 (+4.6) | 77.2 (+6.0) |

# B    IMPLEMENTATION DETAILS

## B.1    NETWORK ARCHITECTURE

Phoenix builds upon the SAM Kirillov et al. (2023) architecture with specific modifications optimized for the mask refinement task. Our architecture leverages key components from SAM while incorporating specialized elements for refinement.

The encoder component consists of the pre-trained ViT-H Dosovitskiy et al. (2021) encoder from SAM, which remains frozen during training to maintain computational efficiency. This encoder generates image embeddings $\mathbf{E}_{img}$ with dimensions $256 \times 64 \times 64$ given an input image of $1024 \times 1024$ resolution.

The decoder takes the noisy input mask as a dense prompt through SAM's prompt encoder. The mask prompt is processed at $4\times$ lower resolution than the input image, then downscaled an additional $4\times$ using two convolutional layers with GELU Hendrycks & Gimpel (2016) activations and layer normalization Ba et al. (2016), ultimately mapping to a 256-dimensional embedding. These dense prompt embeddings are added element-wise with the image embeddings $\mathbf{E}_{img}$ to incorporate mask information.

Additionally, we derive point and box coordinates from the noisy input mask to obtain visual prompt embeddings $\mathbf{E}_v$. For points, we extract positional encodings summed with learnable foreground/background embeddings. For boxes, we encode the top-left and bottom-right corners using positional encodings combined with learned corner-specific embeddings. During the adversarial mask perturbation process, the perturbation embeddings $\mathbf{E}_p$ are concatenated with these visual prompt embeddings, maintaining the standard embedding dimension of 256.

The mask decoder follows SAM's transformer Vaswani et al. (2017) decoder architecture with two layers. Each layer performs self-attention on the tokens, cross-attention from tokens to image embeddings, MLP updates to tokens, and cross-attention from image embeddings to tokens. This structure allows bidirectional information flow between prompts and image features. After decoder processing, the updated image embedding is upsampled by $4\times$ with two transposed convolutional layers. The final mask prediction uses a dot product between the upscaled image embedding with dimensions $32 \times 256 \times 256$ and the output token embeddings.

For the Contrastive Mask Refinement Learning (CMRL), we implement a projector network $g$ that transforms the upsampled image embeddings $\mathbf{F}$ into a space optimized for contrastive learning. This projector consists of 3-layer MLP blocks, generating the projection feature maps $\mathbf{P} = g(\mathbf{F})$ with dimensions $32 \times 256 \times 256$.

## B.2 Training Protocol

Our Phoenix is implemented using the Pytorch framework Paszke (2019). Phoenix is trained using the AdamW optimizer Loshchilov & Hutter (2017) with $\beta_1 = 0.9$, $\beta_2 = 0.999$, an initial learning rate of $1 \times 10^{-4}$, linear warmup for 500 iterations, and cosine decay scheduling. Training proceeds with a total batch size of 16 (2 samples per GPU) on 8 V100 GPUs, weight decay of $5 \times 10^{-4}$, and gradient clipping with a maximum norm value of 0.1.

For each adversarial mask perturbation, we initialize perturbation embeddings $\mathbf{E}_p$ with zeros and add random noise (normal distribution with a mean of 0.0 and a standard deviation of 0.1).

To prevent over-reliance on visual prompts, we randomly omit them with a 30% probability during training by guiding only the noisy mask into the model without the visual prompts. We apply data augmentation, including color jittering and random horizontal flipping, to increase training diversity.

We will release the code upon acceptance.

## B.3 Inference Strategy

Phoenix inherits SAM's multimask output capability, generating multiple refined mask candidates and selecting the one with the highest IoU prediction score. This approach leverages the predictive confidence of the model to identify the most accurate refinement.

Following the strategy introduced in Lin et al. (2025), our model implements cascade self-refinement during inference, where each refinement output serves as input for subsequent iterations. This process, repeated for multiple steps, progressively enhances mask quality through iterative improvement. We empirically determine the optimal number of refinement iterations through detailed analysis, as shown in Figure 5a. Our results indicate that 5 iterations provide the most favorable balance between refinement performance and computational efficiency. Each individual refinement step requires only 6 ms for decoder inference on a V100 GPU, resulting in an additional 24 ms total processing time for the complete refinement cascade. This minimal computational overhead maintains highly efficient inference while substantially improving mask quality. In addition, we enhance model training by incorporating this cascade strategy as a curriculum learning mechanism, gradually transitioning from difficult noisy masks to easier self-refined versions.

## B.4 Details of Self-Supervised Mask Refinement

Our self-supervised approach eliminates the need for ground-truth annotations through a simple yet effective pseudo-supervision strategy, as described in Section A.1. For the pseudo-target generation, we leverage SAM's automatic mask generation mode by prompting it with a fixed grid of $4 \times 4$ points across the image. From the resulting mask candidates, we filter out low-confidence predictions and randomly select one high-confidence mask to serve as our pseudo-target. This approach utilizes SAM's strong zero-shot capabilities to create high-quality reference masks without human annotation. Once the pseudo-targets are generated, we apply the same adversarial perturbation process and training procedure as in our fully-supervised setting. The model learns to refine synthetically perturbed versions of these pseudo-targets back to their original state, effectively transferring SAM's segmentation capabilities into our efficient refinement architecture.

## B.5 Details of Semantic Correlation Analysis

To quantitatively evaluate the fundamental differences between adversarial and morphological noise patterns, we developed a semantic correlation analysis framework. This analysis aims to measure and characterize how different types of noise patterns relate to semantic image features, providing insights into why our adversarial approach generates more realistic and challenging training examples. This analysis was performed on the LVIS validation set. For each image-mask pair, we compute the Pearson correlation Pearson (1895) between the spatial distribution of noise and semantic features:

$$\text{Corr}(S, N) = \frac{\sum_{x,y}(S(x,y) - \bar{S})(N(x,y) - \bar{N})}{\sqrt{\sum_{x,y}(S(x,y) - \bar{S})^2 \sum_{x,y}(N(x,y) - \bar{N})^2}} \tag{5}$$

where $S(x, y)$ represents semantic feature strength (derived from edge and texture maps extracted using Canny edge detection Canny (1986) and Laplacian of Gaussian filtering Marr & Hildreth (1980)) and $N(x, y)$ represents noise magnitude at position $(x, y)$.

Figures 2b and 2c in the main paper provided the distribution of semantic correlation values for both noise types. Morphological noise exhibits a compact, near-symmetric distribution centered close to zero (-0.2 to 0.3), indicating little relationship between noise placement and semantic features. This confirms our hypothesis that conventional morphological operations create perturbations based primarily on geometric constraints rather than semantic understanding. In contrast, our adversarial noise shows a substantially broader distribution (-0.6 to 0.8) extending into both positive and negative correlation regions, demonstrating that it captures a diverse spectrum of semantic relationships.

This broader distribution of adversarial noise is highly beneficial for mask refinement learning for several reasons: (1) it provides comprehensive error coverage across both semantically meaningful regions and homogeneous areas, (2) it creates challenging examples at complex boundaries through positive correlations while generating hard negatives in seemingly simple regions through negative correlations, (3) it better mirrors the diverse error patterns produced by real segmentation models, which make mistakes with varying semantic correlations depending on the context.

### B.6 DETAILS OF CONTRASTIVE MASK REFINEMENT LEARNING (CMRL)

This section provides practical implementation details for CMRL, addressing how region masks and projection features are computed and batched in practice. Algorithm 2 presents the PyTorch-style pseudo-code.

**Region Mask Computation.** Given three binary masks, *i.e.*, target $\mathcal{M}_t$ (ground truth), noisy $\mathcal{M}_n$ (input), and refined $\mathcal{M}_r$ (current prediction), we compute six region masks through logical operations. Each mask is binarized using a 0.5 threshold. The six regions are then derived: $\mathcal{T}_{fg}$ identifies pixels classified as foreground in all three masks, $\mathcal{F}_{fg}$ captures false negatives that remain uncorrected (true foreground but predicted as background in both noisy and refined masks), and $\mathcal{S}_{fg}$ represents successful corrections (true foreground, initially misclassified, but corrected in the refined mask). The same logic applies to background regions with subscript $bg$. This categorization creates a pixel-level curriculum where each spatial position belongs to exactly one of the six mutually exclusive regions.

**Feature Projection and Sampling.** The upsampled image embeddings $\mathbf{F} \in \mathbb{R}^{c \times h \times w}$ are processed through projector $g$ to obtain $\mathbf{p} = g(\mathbf{F}) \in \mathbb{R}^{c \times h \times w}$. where $c = 32$, $h = 256$, and $w = 256$ in our implementation. We then apply L2 normalization along the channel dimension, converting dot products into cosine similarities. For computational efficiency, we sample up to 256 pixels from each region using uniform random sampling. This yields feature vectors $\mathbf{p}_i \in \mathbb{R}^c$ for each sampled position $i$. The loss components are computed separately for each image in the batch and averaged.

**Role of Each Loss Component.** The three loss components serve distinct but complementary purposes. The intra-class loss encourages features from failure regions to align with features from correct regions of the same semantic class, achieving within-class consistency by pulling failure foreground features ($\mathcal{F}_{fg}$) toward correct foreground features ($\mathcal{S}_{fg} \cup \mathcal{T}_{fg}$) through the InfoNCE objective Oord et al. (2018). The inter-class loss enforces separation between foreground and background features by pushing failure foreground features away from all background regions ($\mathcal{F}_{bg} \cup \mathcal{S}_{bg} \cup \mathcal{T}_{bg}$), creating clearer decision boundaries in the feature space. The self-improvement loss enables the model to learn from its own successful corrections by guiding current failures ($\mathcal{F}_{fg} \cup \mathcal{F}_{bg}$) toward successfully refined regions ($\mathcal{S}_{fg} \cup \mathcal{S}_{bg}$), creating a bootstrapping mechanism where the model progressively improves by learning from regions it has already corrected within the same image.

**Projector Architecture and Ablation.** The projector $g$ consists of three 1×1 convolutional layers with LayerNorm and GELU activations between layers. Table 8g suggests the architecture choice of the projector. Without projection (identity mapping), performance is limited to 27.1% AP[1]. Single-layer and two-layer projectors achieve 27.9% and 28.5% respectively, while our three-layer design reaches optimal performance at 28.7%. Adding a fourth layer provides no benefit (28.6%), indicating that three layers provide sufficient capacity.

**Algorithm 2** PyTorch-style Pseudo-code for Contrastive Mask Refinement Learning

```
# Input: image features F (Bx32x256x256),
# masks M_t (GT), M_n (Noisy Mask), M_r (Refined Mask) (Bx1xHxW)

# Step 1: Project and normalize features
projector = MLP_layer(in_channels=32, out_channels=32, num_layers=3)
P = F.normalize(projector(F), p=2, dim=1)  # (Bx32x256x256)

# Step 2: Define six region masks (binarize and detach)
M_t_bin = (M_t > 0.5).float().detach()
M_n_bin = (M_n > 0.5).float().detach()
M_r_bin = (M_r > 0.5).float().detach()

T_fg = (M_t_bin==1) & (M_n_bin==1) & (M_r_bin==1) # True positive
T_bg = (M_t_bin==0) & (M_n_bin==0) & (M_r_bin==0) # True negative
S_fg = (M_t_bin==1) & (M_n_bin==0) & (M_r_bin==1) # Success FN->TP
S_bg = (M_t_bin==0) & (M_n_bin==1) & (M_r_bin==0) # Success FP->TN
F_fg = (M_t_bin==1) & (M_n_bin==0) & (M_r_bin==0) # Failure: uncorrected FN
F_bg = (M_t_bin==0) & (M_n_bin==1) & (M_r_bin==1) # Failure: uncorrected FP

# Step 3: Compute losses per batch item
for b in range(B):
    feat = P[b].view(C, -1).T  # (HxW, C) - flattened normalized features

    # Sample pixels from each region (max 256 samples per region)
    anchor_fg = sample_pixels(feat, F_fg[b], num=256)  # Failure foreground
    anchor_bg = sample_pixels(feat, F_bg[b], num=256)  # Failure background
    pos_fg = sample_pixels(feat, T_fg[b] | S_fg[b], num=256) #Correct foreground
    pos_bg = sample_pixels(feat, T_bg[b] | S_bg[b], num=256) #Correct background

    # Intra-class: Pull failures toward correct same-class features (InfoNCE)
    # For foreground: anchor_fg -> pos_fg
    sim_pos = (anchor_fg @ pos_fg.T) / tau  # (NxM)
    sim_all = (anchor_fg @ feat.T) / tau    # (NxHxW)
    L_intra_fg = -mean(logsumexp(sim_pos, dim=1) - logsumexp(sim_all, dim=1))
    # Similarly for background: L_intra_bg

    # Inter-class: Push failures away from opposite-class features
    # For foreground failures -> background regions
    sim_opposite = (anchor_fg @ pos_bg.T) / tau
    L_inter_fg = mean(log(1 + exp(sim_opposite).sum(dim=1)))
    # Similarly for background: L_inter_bg

    # Self-improvement: Guide failures toward success regions
    success_feat = sample_pixels(feat, S_fg[b] | S_bg[b], num=512)
    failure_feat = sample_pixels(feat, F_fg[b] | F_bg[b], num=512)
    sim_success = (failure_feat @ success_feat.T) / tau
    sim_all = (failure_feat @ feat.T) / tau
    L_self = -mean(logsumexp(sim_success, dim=1) - logsumexp(sim_all, dim=1))

# Step 4: Combine losses
L_CMRL = 0.4*L_intra + 0.4*L_inter + 0.2*L_self
```

## C  IN-DEPTH ANALYSIS OF PHOENIX COMPONENTS

To provide a comprehensive understanding of Phoenix's design decisions and component contributions, we conduct detailed analyses that extend beyond the main paper's ablation studies. Our evaluation protocol maintains consistency with the main paper by employing two distinct noise quality scenarios for instance segmentation: PointWSSIS Kim et al. (2023) with 1% supervision (denoted as $AP^1$) representing challenging refinement scenarios, and Mask R-CNN He et al. (2017) with ResNet-50 He et al. (2016) backbone (denoted as $AP^2$) representing moderate refinement challenges. For fine-grained segmentation analysis, we evaluate performance on DIS-UNet Ronneberger et al. (2015) outputs (denoted as $IoU^1$) representing challenging refinement scenarios and DIS-ISNet Shen et al. (2022) outputs (denoted as $IoU^2$) representing moderate refinement challenges, with results averaged across all DIS test splits Qin et al. (2022).

### C.1 COMPUTATIONAL EFFICIENCY ANALYSIS

Table 8 provides a detailed analysis of Phoenix's computational requirements. Our approach balances high performance with efficiency across both training and inference stages.

**Training Efficiency:** The training process completes in under 10 hours on 8 V100 GPUs, fine-tuning only 4M trainable parameters (0.6% of the full model) and consuming approximately 13.5GB of memory per GPU. This efficiency is achieved by our strategic design choice to fine-tune only the lightweight decoder while keeping the heavy encoder frozen.

In addition, we investigate the computational cost of our adversarial mask perturbation (AMP) process during training. As shown in Table 8a, the AMP Time represents the average time required to generate a single noisy mask through adversarial perturbation. Each perturbation embedding update operation requires approximately 6ms on a V100 GPU, and we found that an average of 4.6 updates are needed to satisfy our IoU threshold requirements for each noisy mask generation. Therefore, the complete adversarial perturbation process takes approximately 27ms ($4.6 \times 6$ms) per mask, which is remarkably efficient considering the semantic complexity of the generated noise patterns.

**Inference Efficiency:** The mask refinement time cost shown in Table 8b represents the total refinement (inference) time for processing the COCO train5K dataset, which contains approximately 5K images and 37K masks. Phoenix demonstrates significant performance advantages while maintaining computational efficiency comparable to SAM-based methods. Both Phoenix and SAMRefiner require 0.6 hours for processing, as they share the same underlying SAM network architecture, while SegRefiner requires 1.4 hours. However, Phoenix achieves substantially higher performance (28.7% AP[1]) compared to both SAMRefiner (21.8% AP[1]) and SegRefiner (14.7% AP[1]), demonstrating superior efficiency in terms of performance per computational cost.

This computational efficiency stems from Phoenix's architectural advantage inherited from SAM, particularly the ability to reuse image embeddings in the lightweight decoder. Once the heavy encoder processes an image to generate embeddings, the lightweight decoder can efficiently refine multiple masks from the same image by reusing these precomputed embeddings. This design is highly advantageous for scenarios where a single image contains multiple masks, such as conventional instance and semantic segmentation tasks, where the computational cost scales primarily with the number of masks rather than images.

#### C.1.1 EFFECT OF IoU THRESHOLDS

Table 8c examines how different IoU threshold ranges during training affect Phoenix's performance. The full range $\mathcal{U}(0.3, 0.9)$ yields the best overall performance, providing exposure to diverse noise levels from severe perturbations (IoU around 0.3) to subtle distortions (IoU around 0.9). Narrower ranges limit the model's exposure to the full spectrum of noise patterns, resulting in reduced performance, particularly on fine-grained segmentation tasks.

#### C.1.2 EFFECT OF GUIDANCE MASKS

Table 8d analyzes how different guidance mask configurations influence refinement performance. The results show that each guidance mask type contributes uniquely to overall performance. Individual expansion or contraction guides provide moderate improvements, while inversion alone performs better than either. Combining expansion and contraction yields strong results (28.5% AP[1] and 46.8% AP[2]), and the full combination achieves optimal performance (28.7% AP[1] and 46.9% AP[2]). This analysis confirms the importance of diverse error patterns for robust refinement performance. By exposing the model to complementary types of errors, we enable it to handle various refinement scenarios effectively.

#### C.1.3 IMPACT OF MORPHOLOGICAL NOISE INTEGRATION

We investigate the integration of classical morphological operations with our adversarial perturbation approach to determine the optimal noise generation strategy for mask refinement training. To this end, we introduce a probability parameter $p_{morph}$ that controls the random replacement of adversarial noise masks with morphological noise masks during training, where $p_{morph} = 0.0$ corresponds to using purely adversarial noise and $p_{morph} = 1.0$ corresponds to using only morphological noise. As

Table 8: **Ablation study** using instance segmentation results on 1% PointWSSIS (AP$^1$) and MRCNN with a ResNet-50 backbone (AP$^2$) and fine-grained segmentation results on DIS-TE1-UNet (IoU$^1$) and DIS-TE1-ISNet (IoU$^2$).

(a) Efficiency of Phoenix

| Specification | Value |
|---|---|
| Training Time | $\leq$ 10h |
| Training Params | 4M (0.6%) |
| GPU Memory Usage | $\approx$ 13.5 GB |
| AMP Time | $\approx$ 27 ms |

(b) Mask refinement time cost

| Method | AP$^1$ | Time (h) |
|---|---|---|
| SegRefiner | 14.7 | 1.4 |
| SAMRefiner | 21.8 | 0.6 |
| Phoenix (ours) | 28.7 | 0.6 |

(c) $\tau$ (IoU Threshold)

| $\tau$ | AP$^1$ | AP$^2$ | IoU$^2$ | IoU$^2$ |
|---|---|---|---|---|
| $\mathcal{U}(0.3, 0.9)$ | 28.7 | 46.9 | 75.7 | 77.1 |
| $\mathcal{U}(0.5, 0.9)$ | 28.1 | 47.0 | 74.7 | 77.2 |
| $\mathcal{U}(0.3, 0.7)$ | 28.8 | 46.4 | 75.9 | 75.8 |
| $\mathcal{U}(0.5, 0.7)$ | 28.3 | 46.6 | 74.9 | 76.9 |

(d) $\mathcal{M}_g$ (Guidance Mask)

| Expansion | Contraction | Inversion | AP$^1$ | AP$^2$ |
|---|---|---|---|---|
| ✓ | ✗ | ✗ | 26.9 | 46.1 |
| ✗ | ✓ | ✗ | 27.1 | 46.2 |
| ✗ | ✗ | ✓ | 28.0 | 46.5 |
| ✓ | ✓ | ✗ | 28.5 | 46.8 |
| ✓ | ✓ | ✓ | **28.7** | **46.9** |

(e) $p_{morph}$

| $p_{morph}$ | AP$^1$ |
|---|---|
| 0.0 | 28.7 |
| 0.1 | 28.6 |
| 0.3 | 27.7 |
| 0.5 | 26.1 |
| 1.0 | 23.8 |

(f) Encoder Study

| Encoder | AP$^1$ | AP$^2$ | VRAM (GB) | GFlops (G) | FPS |
|---|---|---|---|---|---|
| EfficientViT-XL1 | 28.0 | 45.9 | 0.8 | 323 | 45.2 |
| ViT-B | 22.4 | 42.0 | 2.8 | 369 | 12.4 |
| ViT-L | 26.6 | 46.3 | 4.1 | 1313 | 5.6 |
| ViT-H | 28.7 | 46.9 | 4.9 | 2735 | 3.5 |

(g) Projector $g$

| Projector | AP$^1$ | AP$^2$ |
|---|---|---|
| Identity | 27.1 | 46.2 |
| 1-layer MLP | 27.9 | 46.5 |
| 2-layer MLP | 28.5 | 46.7 |
| **3-layer MLP** | **28.7** | **46.9** |
| 4-layer MLP | 28.6 | 46.8 |

shown in the Table 8e, performance consistently degrades with increasing $p_{morph}$ values, declining from 28.7% AP$^1$ when using purely adversarial noise ($p_{morph} = 0.0$) to 23.8% AP$^1$ when relying exclusively on morphological noise ($p_{morph} = 1.0$). This demonstrates that traditional morphological operations are insufficient for generating realistic noise patterns. These results underscore the importance of our semantic-aware adversarial noise modeling, which captures more nuanced and contextually relevant perturbations compared to the geometric transformations provided by classical morphological operations, ultimately leading to more effective mask refinement capabilities.

### C.1.4 EFFECT OF ADVERSARIAL MASK PERTURBATION PARAMETERS

We analyze how various parameters affect our adversarial perturbation process to provide practical guidance for implementation. Figure 5b examines Phoenix's sensitivity to the initial step size ($\alpha_0$) and maximum iteration count ($N$), revealing remarkable stability across a wide range of values with AP$^1$ consistently above 28.0% for $N \geq 5$ and $\alpha_0 \in [0.001, 0.1]$. The optimal configuration is achieved at $\alpha_0 = 0.01$ and $N = 10$, while extreme values (very small $\alpha_0$ or large $\alpha_0$ with small $N$) should be avoided as they lead to insufficient or unrealistic perturbations.

Similarly, Figure 5c shows the impact of perturbation embedding count ($P$), with performance improving substantially as $P$ increases from 1 to 50 (AP$^1$ rising from 26.7% to 29.0%), then plateauing beyond this point. This suggests that $P = 50$ provides an optimal balance between effectiveness and efficiency, capturing the necessary perturbation patterns without excessive computational overhead.

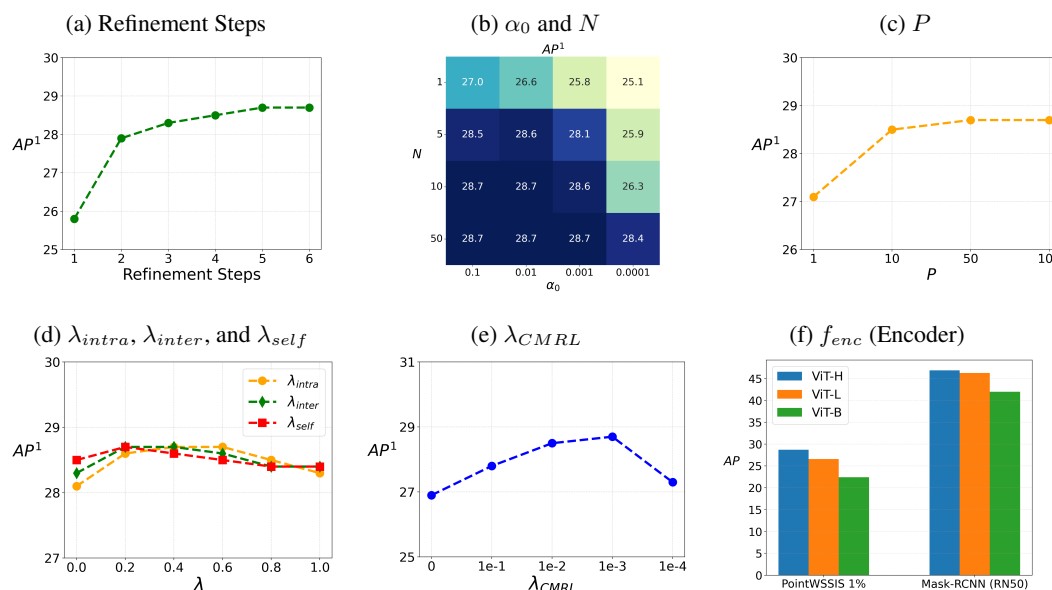

Figure 5: **Ablation study (continue)** using instance segmentation $AP^1$ results.

Through this analysis, we establish that once these parameters (*i.e.*, $\alpha_0$, $N$, and $P$) are set to reasonable values, **the IoU threshold parameter $\tau$ becomes the primary parameter** to control the perturbation intensity without sophisticated parameter tuning.

### C.1.5 EFFECT OF CONTRASTIVE MASK REFINEMENT LEARNING (CMRL) LOSS WEIGHTS

We conduct a comprehensive analysis of how CMRL component weights influence Phoenix's performance to optimize our contrastive learning strategy. The total training loss of Phoenix combines the original SAM loss function with our contrastive loss: $\mathcal{L} = \mathcal{L}_{SAM} + \lambda_{CMRL} \cdot \mathcal{L}_{CMRL}$, where $\mathcal{L}_{SAM}$ is the linear combination of Focal Lin et al. (2017) and Dice Milletari et al. (2016) loss in a 20:1 ratio.

Figure 5d reveals how individual contrastive components ($\lambda_{intra}$ for intra-class consistency, $\lambda_{inter}$ for inter-class contrast, and $\lambda_{self}$ for self-improvement regularization) affect refinement quality. We systematically evaluate each component by fixing our default configuration ($\lambda_{intra} = 0.4$, $\lambda_{inter} = 0.4$, $\lambda_{self} = 0.2$) and varying each weight individually from 0.0 to 1.0. All three components exhibit similar inverted U-shaped performance curves, with effectiveness peaking around 0.4 before gradually declining with higher values. This pattern suggests that while each contrastive component provides valuable learning signals, excessive emphasis on any single component can undermine the primary segmentation objective. The optimal balance places greater emphasis on intra-class consistency and inter-class contrast compared to self-improvement regularization, indicating that feature space organization (establishing clear boundaries between foreground and background while ensuring consistency within each class) is particularly crucial for effective refinement.

Meanwhile, as shown in Figure 5e, the overall CMRL scaling factor $\lambda_{CMRL}$ reaches optimal performance at the relatively small value of $1 \times 10^{-3}$, achieving an $AP^1$ of 28.7%. Performance degrades significantly with higher values (dropping to 27.3% at $\lambda_{CMRL} = 0.1$), confirming that contrastive learning should complement rather than dominate the training process.

These findings provide important practical guidance: while our tri-directional contrastive approach significantly enhances mask refinement, it requires careful integration with conventional segmentation objectives. The sensitivity analysis demonstrates that a light touch with contrastive learning, applying it as a targeted enhancement to traditional segmentation loss, yields the most robust and effective refinement model.

## C.2 EFFECT OF ENCODER ARCHITECTURE

To provide guidance for practical deployment across different computational constraints, we investigate how various ViT Dosovitskiy et al. (2021) encoder architectures impact Phoenix's performance. This analysis is important for understanding the trade-offs between model capacity and refinement quality, helping practitioners select the appropriate configuration for their specific requirements.

During training, we freeze the image encoder and fine-tune only the lightweight decoder with mini-batch size 2 (total batch size 16 with 8 GPUs) using FP16 precision on V100 GPUs. In this setting, we measure the latency and VRAM of the image encoder using mini-batch size 2 inputs ($2{\times}3{\times}1024{\times}1024$). As shown in Table 8f and Figure 5f, we evaluate both standard ViT variants and the lightweight EfficientViT Zhang et al. (2024).

The results demonstrate clear trade-offs between model capacity, performance, and computational efficiency. ViT-H achieves the best refinement quality (28.7% AP[1] and 46.9% AP[2]), leveraging its larger capacity to capture richer image features, but requires substantial computational resources (4.9GB VRAM, 3.5 FPS). ViT-L shows minimal performance degradation while offering improved efficiency. The smaller ViT-B encoder shows more substantial drops (22.4% AP[1] and 42.0% AP[2]) but significantly reduces computational requirements (2.8GB VRAM, 12.4 FPS). Most notably, EfficientViT-XL1 provides an excellent accuracy-speed trade-off with competitive performance (28.0% AP[1] and 45.9% AP[2]) while achieving remarkable efficiency (45.2 FPS and minimal 0.8GB VRAM usage), offering flexible deployment options for different computational constraints. The per-GPU computational requirements are highly efficient across all variants, which is much lower than typical memory-intensive vision models.

## C.3 QUALITATIVE NOISE PATTERN ANALYSIS

A critical component of Phoenix is our adversarial mask perturbation (AMP) mechanism, which generates realistic training noise that closely approximates real-world segmentation failures. Figure 8 provides additional noisy mask samples of our adversarial masks compared to traditional morphological noise masks. Unlike morphological operations that apply predictable geometric transformations directly to mask pixels, our approach generates semantically-aware perturbations by embedding space adversarial attacks. This enables the generation of diverse failure patterns, including contextual confusion, over- or under-segmentation errors, and boundary imprecision.

The visual comparison in Figure 8 reveals that our adversarial masks closely resemble real segmentation failures from production models (1% PointWSSIS for instance segmentation, DIS-UNet for fine-grained segmentation), while morphological masks show limited, uniform patterns. This improved realism directly contributes to Phoenix's superior mask refinement performance.

# D ADDITIONAL QUALITATIVE RESULTS

We present comprehensive qualitative results to demonstrate Phoenix's effectiveness across diverse segmentation tasks and challenging scenarios. These visual comparisons complement our quantitative analyses and provide insights into the specific types of improvements Phoenix achieves.

Figure 9 provides extensive qualitative comparisons on the LVIS dataset for instance segmentation tasks, showing Phoenix's performance relative to SegRefiner and SAMRefiner across diverse object categories. The examples span various scales, from small objects to large objects, demonstrating Phoenix's robust refinement capabilities. Particularly notable are the improvements in handling complex poses and occlusions, as seen in the human figures and animal examples. Phoenix consistently produces more accurate boundary delineations and better preserves object details compared to existing methods.

Figure 10 shows additional qualitative results on the DIS benchmark, comparing Phoenix with SegRefiner and SAMRefiner on fine-grained segmentation tasks. The examples highlight Phoenix's superior ability to handle complex structures and intricate boundaries. Namely, Phoenix successfully refines the delicate feather structures of various objects, preserving fine details that other methods struggle to capture. The bicycle examples demonstrate Phoenix's effectiveness in handling complex mechanical structures with thin components like spokes and cables. For architectural elements

like windmills and playground equipment, Phoenix maintains structural integrity while refining boundary precision. The dinosaur skeleton example illustrates Phoenix's capability with highly detailed, branching structures that require precise boundary delineation.

These qualitative results consistently demonstrate Phoenix's superior refinement quality across diverse scenarios, object categories, and segmentation tasks. The visual improvements align with our quantitative findings, confirming Phoenix's effectiveness as a general-purpose mask refinement approach that can enhance segmentation quality across various application domains.

## E    FAILURE CASE ANALYSIS

Despite Phoenix's strong overall performance, we conduct a comprehensive analysis of challenging scenarios where our method faces limitations. Figure 11 presents representative failure cases organized into three categories that highlight the current boundaries of our approach and provide insights for future improvements.

**Ambiguous Target Objects:** This failure mode occurs when the target object itself is inherently difficult to distinguish or when multiple similar objects create boundary confusion. In Figure 11a first row, the unclear target object mask between two occluded giraffes presents an ambiguous segmentation scenario where even determining the correct target is challenging. The spatial overlap and visual similarity between the two giraffes make it difficult to establish which object should be segmented. Similarly, in Figure 11b second row, the noisy mask contains multiple objects, creating confusion about whether to remove or maintain the pot in the final segmentation. Such ambiguous scenarios arise when the input contains insufficient contextual information to resolve target object identity, making the refinement task inherently ill-defined.

**Totally Mislocalized Input Noisy Masks:** This represents the most severe failure mode where initial noisy masks are completely spatially displaced from the actual target objects. Examples include Figure 11a second and third rows, and Figure 11b first row, where the noisy masks bear no spatial correspondence to the ground-truth mask locations. In these cases, the refinement problem becomes fundamentally unsolvable because there is no meaningful overlap or spatial relationship between the input mask and the actual object boundaries. This failure mode reveals the inherent limitation of mask refinement approaches that depend critically on the initial quality of noisy masks. These failures underscore that refinement-based methods have fundamental prerequisites regarding input mask quality and cannot recover from arbitrary initialization errors.

**Class-Agnostic Refinement Limitations:** Phoenix's class-agnostic design, while enabling broad generalization, creates limitations when semantic understanding is required for proper refinement. Figure 11c demonstrates cases where Phoenix cannot identify or correct misclassified masks. In the first row, Phoenix cannot separate the person and bike from the merged mask because it lacks semantic understanding to distinguish between different object classes that have been incorrectly combined. The method treats the merged region as a single entity and refines its boundaries accordingly, without recognizing that it should be decomposed into separate semantic categories. In the second row, Phoenix cannot refine the misclassified region where the sail is incorrectly labeled as a person. While Phoenix successfully improves the mask's spatial quality and boundary precision, it maintains the fundamental semantic error because our architecture focuses exclusively on visual boundary refinement without incorporating class-conditional reasoning.

**Future Research Directions:** These failure modes directly connect to the research directions identified in our main paper analysis.

(1) For ambiguous target objects, the integration of visual cues provides a promising solution by guiding the target object to be segmented. As demonstrated in Table 4f of our main paper, incorporating visual prompts (points or boxes) that provide explicit target object information achieves remarkable performance improvements. This approach directly addresses the target specification challenge by providing clear geometric guidance about which object should be the focus of refinement, effectively resolving ambiguity in multi-object scenarios like the occluded giraffes or mixed object cases.

(2) For totally mislocalized masks, visual prompts can also provide spatial anchoring, though the effectiveness depends on the degree of misalignment. When combined with additional spatial reasoning mechanisms, visual cues could help establish the correct target location even when initial masks are severely displaced.

(3) For class-agnostic refinement limitations, our main paper identifies incorporating open vocabulary models Ghiasi et al. (2022); Liang et al. (2023) or text embeddings Li et al. (2023b); Radford et al. (2021) as a promising research direction to handle misclassified object masks. Since Phoenix operates in a class-agnostic manner and cannot correct semantic class errors inherent from real models, integrating vision-language understanding could enable the system to recognize and correct semantic misclassifications while maintaining spatial refinement capabilities. This multimodal extension would allow Phoenix to leverage both visual boundary information and semantic understanding, potentially resolving the sail-as-person misclassification by incorporating textual or semantic priors that distinguish between different object categories.

These insights highlight both the current capabilities and future potential of mask refinement systems, pointing toward more robust, multimodal approaches that can handle increasingly diverse and challenging segmentation scenarios through the integration of visual prompts for target specification and semantic understanding for class-aware refinement.

## F  BROADER IMPACTS

Phoenix offers significant potential to benefit various domains by improving segmentation quality across a wide range of applications. High-quality segmentation is foundational to many computer vision tasks, and our refinement approach provides substantial improvements without requiring architectural changes to base models or extensive retraining. The demonstrated ability to enhance both existing annotations and model outputs suggests Phoenix could reduce manual effort in dataset creation while improving the performance of downstream applications that rely on precise segmentation. By focusing on a refinement paradigm that builds upon existing segmentation methods, Phoenix complements rather than replaces current approaches, allowing for integration into established workflows.

While any advanced technology carries some responsibility for appropriate use, we have designed Phoenix to be broadly applicable to beneficial applications across diverse domains. We are committed to making this technology available to the research community upon publication to encourage further innovation and refinement of segmentation capabilities.

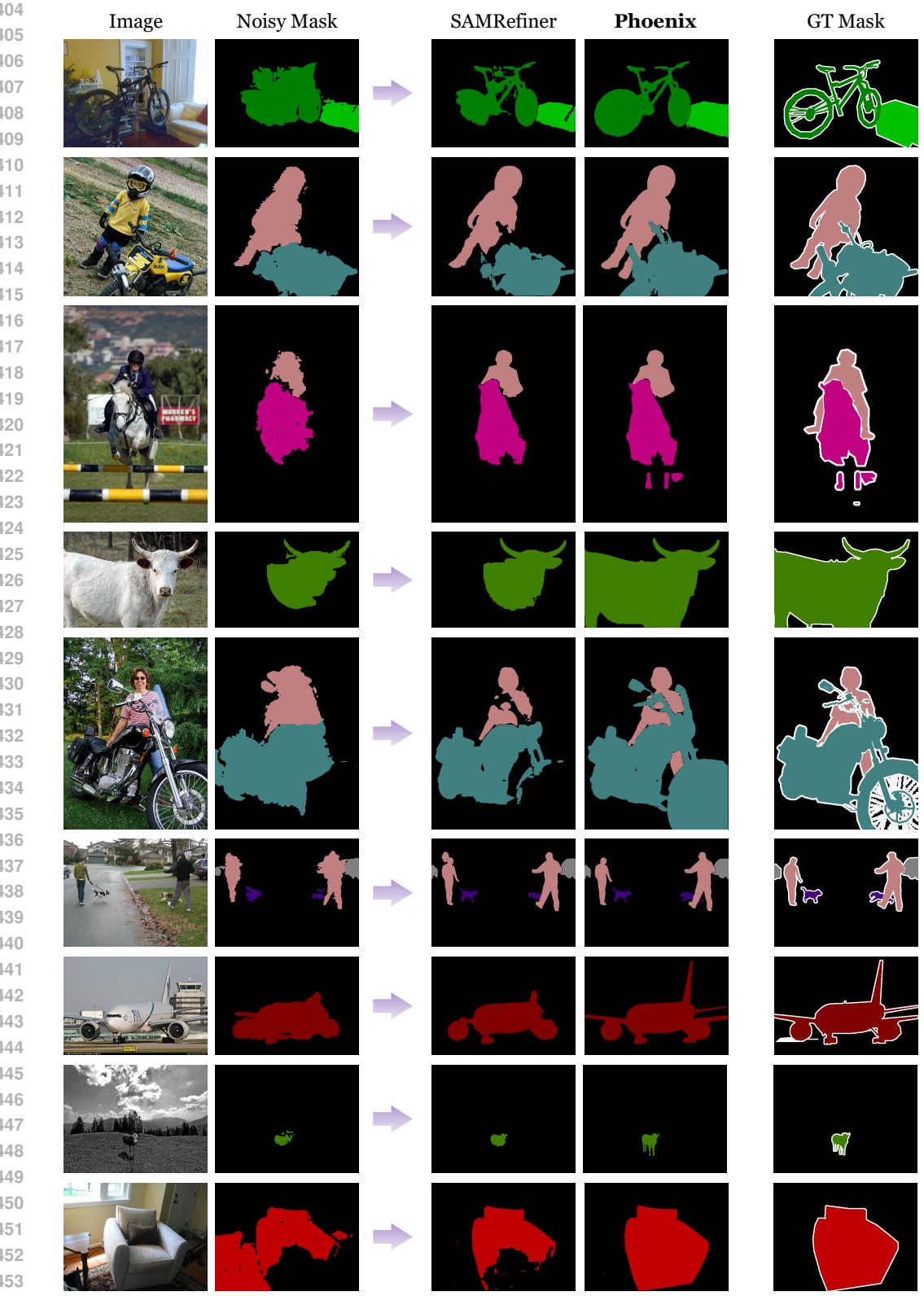

Figure 6: **Qualitative results for semantic segmentation refinement**. Examples show Phoenix's effectiveness in refining coarse semantic masks across diverse scene categories. The method successfully corrects both over-segmentation and under-segmentation errors while maintaining semantic consistency, particularly excelling at complex boundary regions and multi-object scenarios.

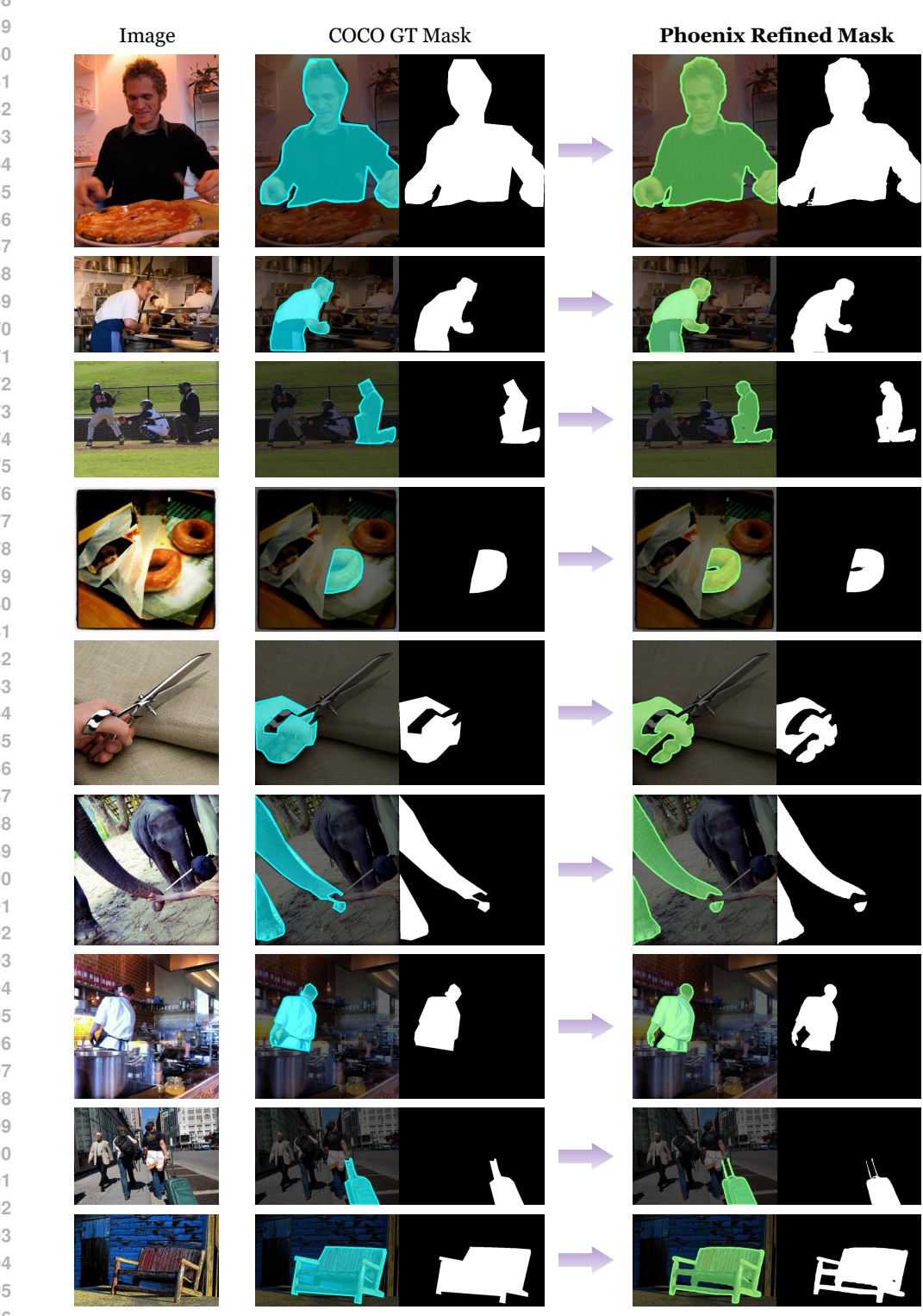

Figure 7: **Qualitative results for human annotation refinement** for COCO 2017 *val* dataset. The left columns show the original COCO ground truth masks. The right columns display Phoenix-refined masks that achieve better alignment with high-quality mask annotations. Phoenix effectively corrects annotation imperfections, particularly improving boundary precision and handling of fine details that are often missed in standard polygon-based annotation workflows.

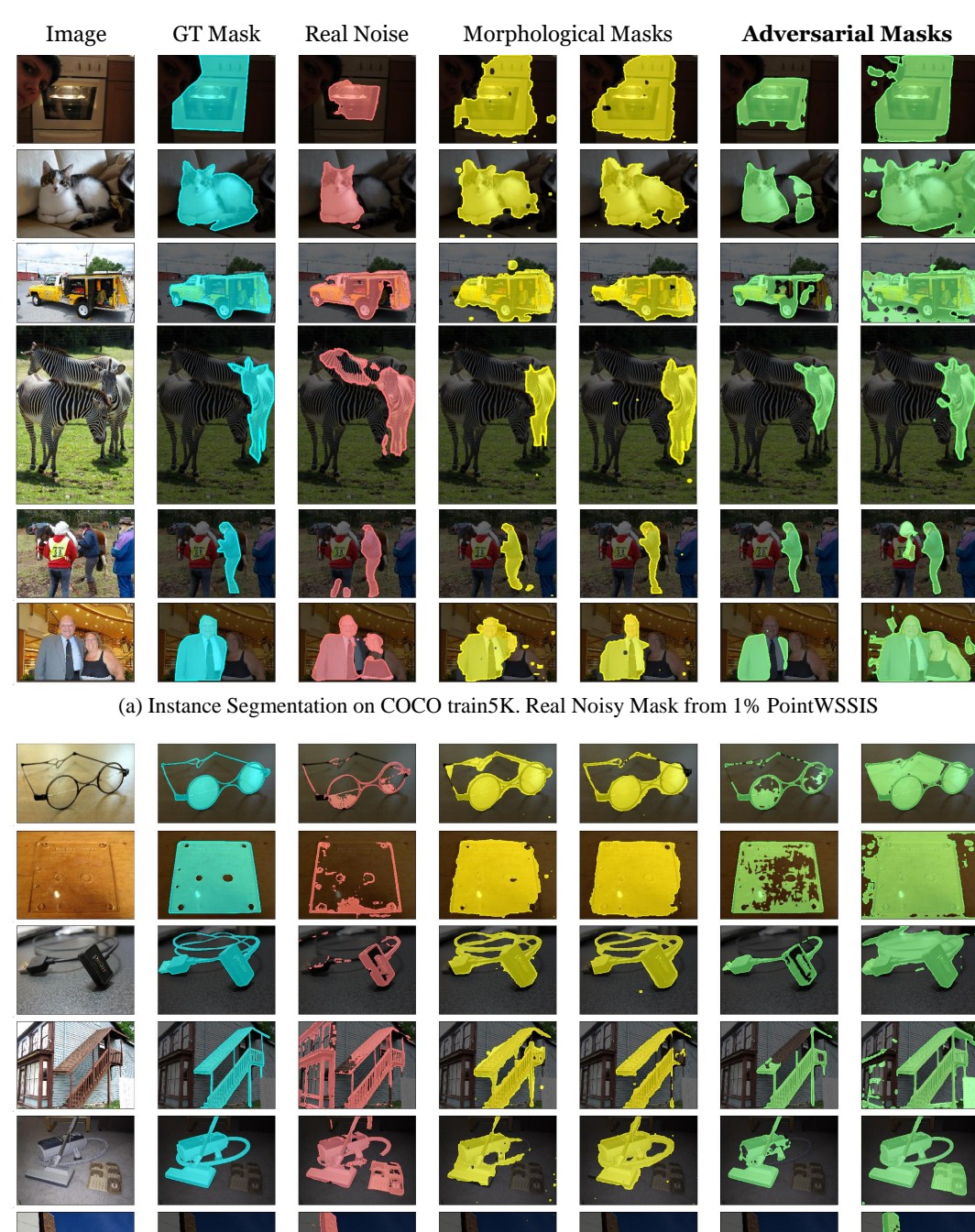

(a) Instance Segmentation on COCO train5K. Real Noisy Mask from 1% PointWSSIS

(b) Fine Segmentation on DIS5K. Real Noisy Mask from DIS-UNet

Figure 8: **Noise pattern analysis** demonstrating the superiority of adversarial mask perturbation over morphological methods. Our approach produces diverse, realistic failure patterns, including contextual errors, segmentation inconsistencies, and boundary imprecision that closely match real segmentation model failures, leading to improved refinement performance.

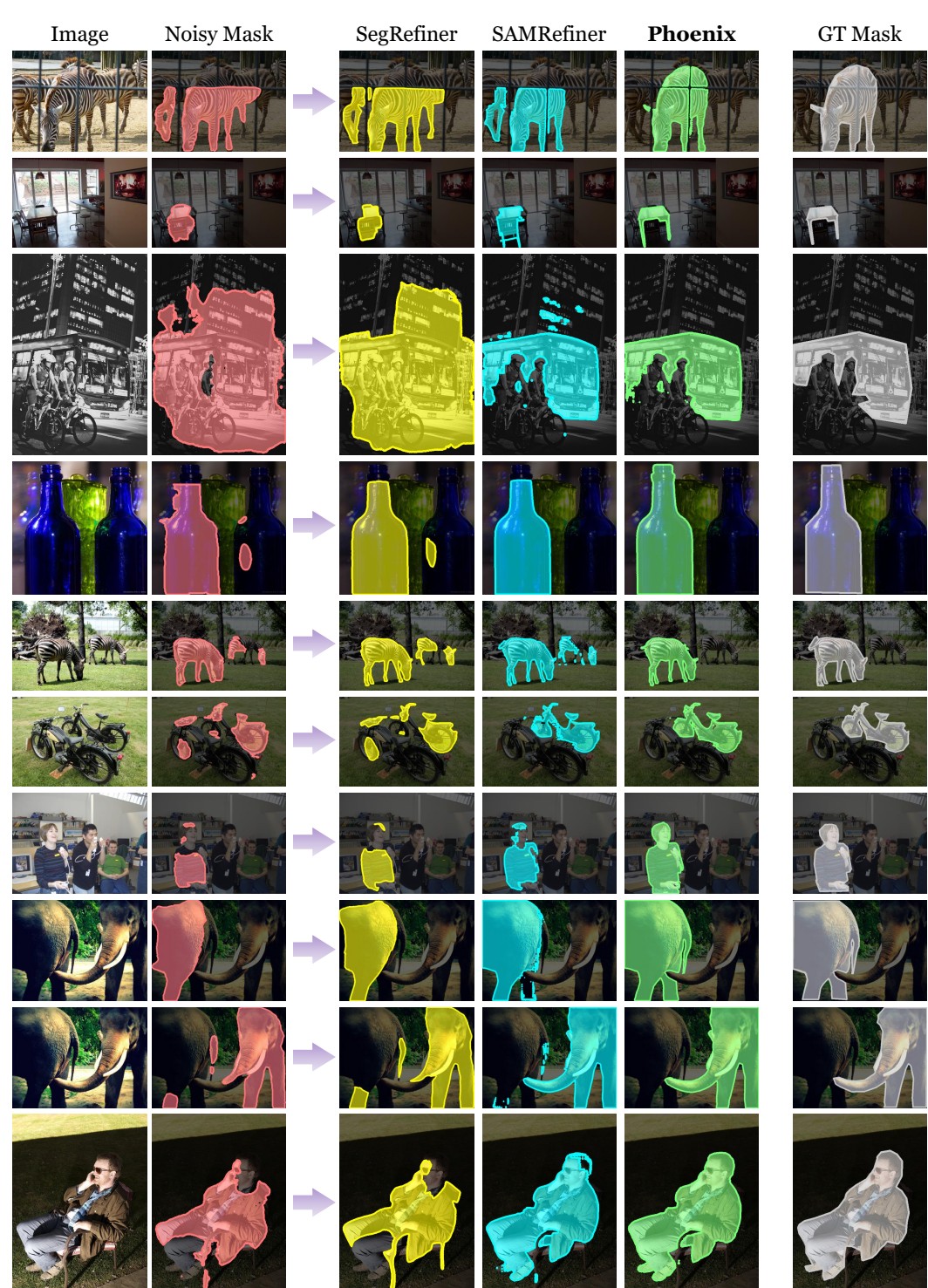

Figure 9: **Additional qualitative results for instance segmentation refinement.** Each row shows progression from noisy input masks through different refinement methods (SegRefiner Wang et al. (2023), SAMRefiner Lin et al. (2025)) to Phoenix's output and ground truth. Phoenix consistently produces more accurate boundary delineation and better handling of complex object structures across diverse object categories.

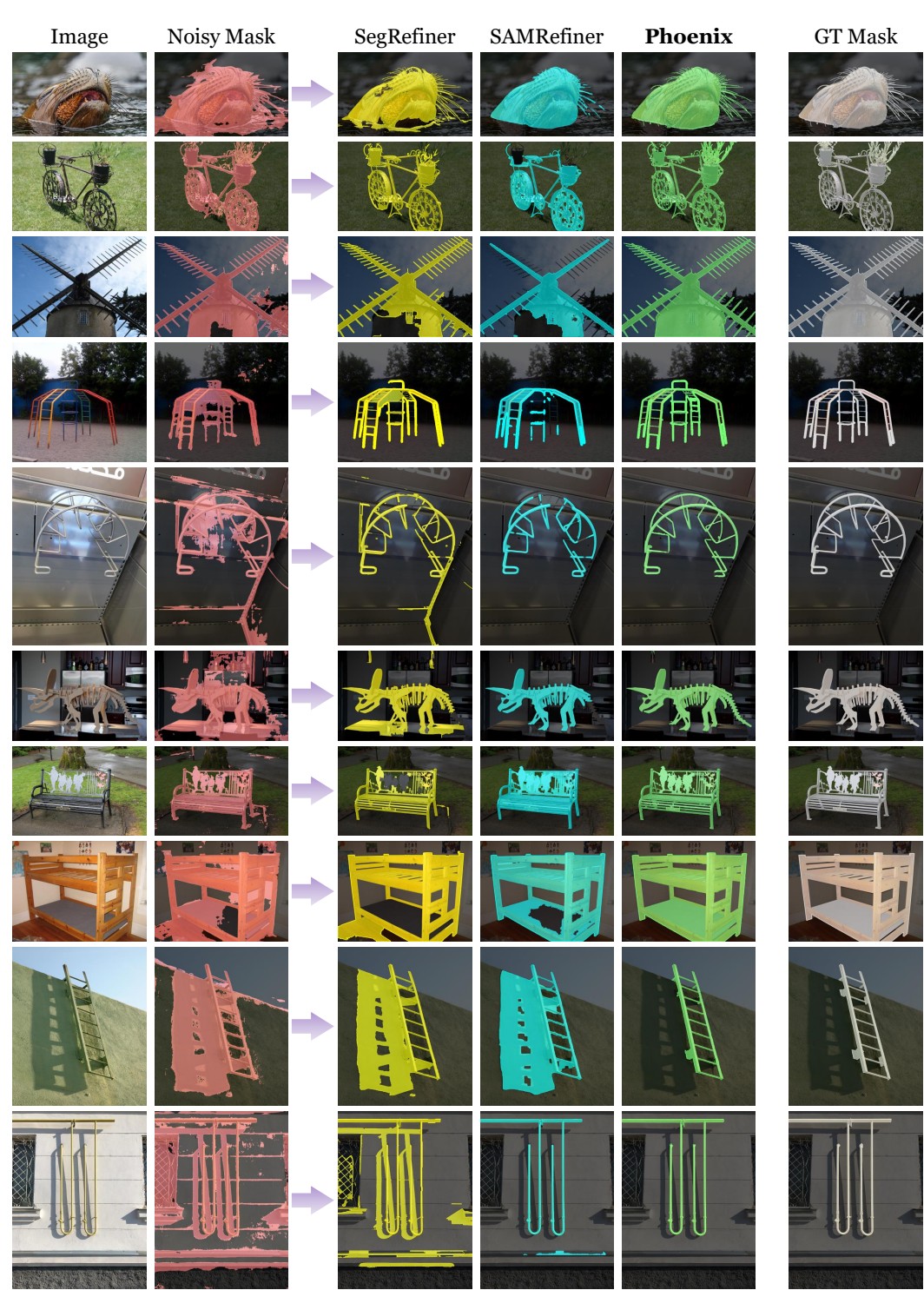

Figure 10: **Additional qualitative results for fine-grained segmentation refinement**. Examples demonstrate Phoenix's capability to handle intricate object boundaries and thin structures across various categories. Phoenix shows superior boundary precision compared to baseline methods, particularly for objects with complex geometric features.

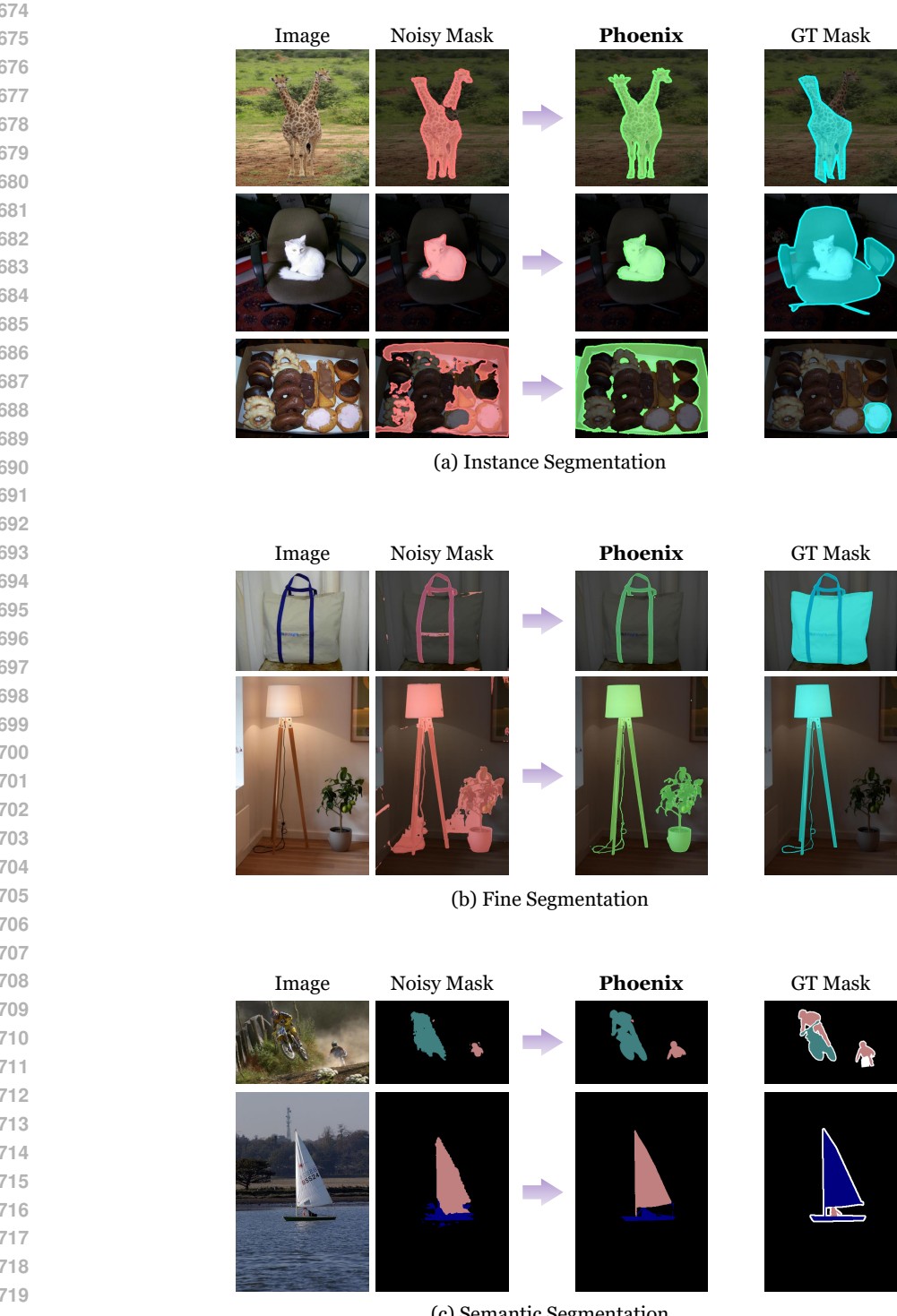

(a) Instance Segmentation

(b) Fine Segmentation

(c) Semantic Segmentation

Figure 11: **Failure case analysis across different segmentation tasks**. (a) Instance segmentation failures include heavily occluded objects (giraffe), completely mislocalized masks (cat and chair), and ambiguous multi-object boundaries (food items). (b) Fine-grained segmentation failures involve complex geometric structures and spatial misalignment issues. (c) Semantic segmentation failures demonstrate class-agnostic limitations where Phoenix refines mask quality but cannot correct semantic misclassifications (sail classified as person). These cases highlight current method boundaries and inform future research directions.

