# OpenReview forum: "Learning from Adversity: Semantic-Aware Mask Refinement through Adversarial Perturbation"
_ICLR.cc/2026/Conference — Submitted to ICLR 2026_

### Official Review · Reviewer_bEY1 · 2025-10-17

**Soundness:** 4
**Presentation:** 3
**Contribution:** 3
**Rating:** 8
**Confidence:** 3

**Summary:**

Although recent advances have greatly improved image segmentation, even state-of-the-art models still produce masks with inaccurate boundaries, semantic inconsistencies, and structural artifacts. While mask refinement techniques aim to address these limitations, existing approaches typically rely on simplistic synthetic noise that fails to represent the complex error patterns found in real segmentation outputs.
The authors present Phoenix, a novel framework that combines adversarial and contrastive learning for realistic mask refinement. Phoenix introduces two core components: (1) Adversarial Mask Perturbation, which employs embedding-level attacks to generate semantically meaningful noise resembling real segmentation errors, and (2) Contrastive Mask Refinement Learning, a tri-directional contrastive formulation that enforces feature consistency within semantic regions while maintaining clear separation across classes.

**Strengths:**

- The authors address an interesting and important topic.
- I appreciate that the authors go into more detail about the limitations of other papers in section 3.
- The method is well described and motivated - everything is easy to follow.
- The approach of adaptive, threshold-controlled noise generation is useful.
- The datasets, evaluation metrics, and models were appropriately selected.
- The results (Tables 2 and 3) are very good compared to SOTA.
- The authors have conducted a large number of ablation studies and made good comparisons. The ablations address each component of the presented method.
- In the appendix, more complex datasets and zero-shot approaches were tested - so great additional results.
- Implementation details are presented, and the authors intend to make code available at a later date.
- The limitations are reflected in failure cases, and based on these and their findings, the authors outline the next possible steps.

**Weaknesses:**

- The related work section seems superficial to me, as if the authors are only focusing on the works they use as baselines.


Remarks:
- The theoretical analysis of semantic distribution is a good idea, but I find the section insufficiently explained and would therefore perhaps only explain it in the appendix.
- I think details for the point annotations are worth adding.
- A brief explanation of the challenging DIS task would be helpful.
- The order of the figures in comparison to when they are referenced in the text is somewhat confusing.

**Questions:**

- Regarding the novelty, are you the first to use adversarial perturbation for mask refinement?
- Why were the NB and PointWSSIS methods chosen as baseline models for (weakly) semi-supervised?

---

> ### Author Response · Authors · 2025-11-19
> **Official Comment by Authors (1/1)**
>
> We sincerely appreciate Reviewer bEY1 for the thorough review and strong positive assessment of our work. We greatly appreciate your recognition of (1) our well-motivated and clearly described method, (2) comprehensive experiments with extensive ablations addressing each component, (3) strong results compared to SOTA, and (4) thorough analysis including zero-shot generalization and failure cases. We have carefully addressed all your suggestions below.
>
> ---
> ## W1: Related Work Section
> We appreciate the constructive feedback. We have expanded the related work with two new subsections:
> - "Adversarial Learning": Covers adversarial learning and positions our constructive use of adversarial perturbation
> - "Contrastive Learning": Discusses contrastive methods in dense prediction tasks and highlights our novel tri-directional framework for refinement
> - These additions provide broader context while maintaining detailed analysis of directly comparable methods.
>
> ---
> ## R1: Semantic Correlation Analysis (Section 3.3)
> Following the suggestion, we have added a brief introduction at Section 3.3 explaining that the analysis computes Pearson correlation between noise spatial distribution and semantic features (edge detection and texture maps), with a reference to Appendix B.5 for detailed methodology.
>
> ---
> ## R2: Point annotation details
> In PointWSSIS (Kim et al. 2023), point annotations are single object center points serving as weak supervision. We have clarified this in Table 1 caption, simply. Importantly, our method does not use the point annotations directly, but refines the output masks from PointWSSIS, which was trained with limited full mask annotations and numerous point annotations.
>
> ---
> ## R3: Challenges of DIS task
> We have added clarification in Section 4.1 that the DIS task "demands precise boundary delineation for thin structures, complex topologies, and fine-grained details," making the evaluation challenges immediately clear.
>
> ---
> ## R4: Figure ordering
> We acknowledge this valid concern. Figure 1 is intentionally placed early to provide immediate visual demonstration of Phoenix's effectiveness on the opening page. To address the confusion, we have added forward references in the introduction (e.g., "as shown in Figure 1" and "as shown in Figure 2"). We remain open to reordering if you believe strict sequential ordering would significantly improve readability.
>
> ---
> ## Q1: Novelty of adversarial perturbation
> To the best of our knowledge, we are the first to use adversarial perturbation for mask refinement training data generation. Our novelties: (1) repurposing adversarial attacks as constructive data augmentation, (2) embedding-space perturbation for efficiency, (3) controllable noise via guidance masks and IoU thresholds, (4) semantic-aware noise.
>
> ---
> ## Q2: Why NB and PointWSSIS as baselines?
> NB and PointWSSIS are prone to generating noisy masks due to their challenging supervision settings (weakly or semi-supervised learning). Their performance varies significantly with the amount of full supervision (1%, 5%, 10%), making them ideal for evaluating mask refinement effectiveness across diverse noise conditions ranging from extreme to moderate quality.
> Moreover, we follow the established evaluation protocol from SAMRefiner (Lin et al. ICLR 2025), which uses NB (Wang et al. CVPR 2022) and PointWSSIS (Kim et al. CVPR 2023) as standard benchmarks for mask refinement in (weakly) semi-supervised settings.
> This choice ensures fair comparison with prior refinement work while evaluating performance across diverse noise conditions relevant to label-efficient learning.

---

> > ### Comment · Reviewer_bEY1 · 2025-11-24
> >
> > Thank you for the reponse and clarifications. I'm keeping my positive score.

---

### Official Review · Reviewer_ADPo · 2025-10-31

**Soundness:** 3
**Presentation:** 2
**Contribution:** 3
**Rating:** 4
**Confidence:** 3

**Summary:**

The paper introduces Phoenix, a semantic-aware mask refinement framework that leverages adversarial perturbations and contrastive learning. The method consists of two main modules: (1) AMP, which generates semantically meaningful noise to simulate realistic segmentation errors; and (2) CMRL, a tri-directional contrastive framework aligning ground-truth, noisy, and refined masks.

**Strengths:**

- The idea of using adversarial embedding perturbations for generating realistic segmentation noise is interesting and technically sound.
- The framework is systematically evaluated across multiple segmentation settings, with comprehensive ablations and efficiency analyses showing thoughtful experimentation.

**Weaknesses:**

1. Is the proposed adversarial mask perturbation conceptually similar to augmentation strategies that add positive (or forward) perturbations to enhance robustness? If so, how does Phoenix differ in motivation or mechanism—does it essentially act as a form of task-specific adversarial regularization?
2. Are there cases where adversarial perturbations introduce unrealistic distortions that negatively affect training stability? If so, how are such cases identified or mitigated during training?

**Questions:**

1. Is the proposed **adversarial mask perturbation** similar to these augmentation strategies that add positive perturbations to improve robustness? Is Phoenix effectively a form of task-specific adversarial regularization?
2. Are there cases where adversarial perturbations may introduce unrealistic distortions that harm training stability? If so, how are such cases handled or filtered?

---

> ### Author Response · Authors · 2025-11-19
> **Official Comment by Authors (1/2)**
>
> We sincerely appreciate the reviewer for recognizing the technical soundness of our adversarial embedding perturbation approach and the comprehensive evaluation. We address your concerns below.
>
> ---
> ## W1 & Q1: Difference between augmentation strategies and AMP
> We appreciate this important conceptual question. The term "augmentation strategies that add positive (or forward) perturbations" could refer to different augmentation strategies, so we address both potential interpretations below. If neither interpretation captures your concern, we would gladly clarify during this discussion period.
>
> **(A) Distinction from Adversarial Learning Methods**
>
> - Conventional Adversarial Learning (e.g., Madry et al. 2018, Goodfellow et al. 2015):
>   - Purpose: Makes models robust against adversarial attacks at test time
>   - Mechanism: Creates a min-max game where perturbations attack the model being trained, and the model learns to defend against these attacks
>   - Training: Model is directly exposed to adversarial examples during training: $max_{\delta} \text{Loss}(\text{Model}(x+\delta), y)$
>   - Test time: Model handles adversarial attacks
>
> - Ours (Adversarial Mask Perturbation):
>   - Purpose: Generates realistic training data that mimics real segmentation errors
>   - Mechanism: Uses adversarial optimization as a _tool_ to generate noisy masks on a frozen pretrained decoder, completely separate from the refinement model
>   - Training: Two separated phases: (1) Generate noisy masks via adversarial perturbation on frozen decoder, (2) Train refinement model on pre-generated (noisy, clean) mask pairs
>   - Test time: Refiner handles real segmentation errors, NOT adversarial attacks
>
> - Key differences:
>   1.  Different optimization targets: We optimize perturbations on a frozen pretrained decoder, not on the refinement model being trained. The perturbation embeddings ($E_p$) are NOT part of the refinement model's parameters.
>   2.  Semantic-aware generation: Our theoretical analysis (Section 3.3) shows adversarial perturbations concentrate in high-uncertainty regions. This creates noise mimicking where real segmentation models struggle, not adversarial attacks designed to fool the model.
>   3.  Separated paradigm: Adversarial training couples perturbation generation with model training. Phoenix completely separates data generation (using frozen decoder) from model training (refining on generated pairs).
>   4. Phoenix is "adversarial data synthesis" rather than "adversarial regularization."
>
> **(B) Distinction from Spatial Mask Augmentation Strategies**
> - Traditional Spatial Mask Augmentation (e.g., morphological operations in SegRefiner):
>   - Method: Applies fixed geometric transformations (dilation, erosion, boundary perturbations)
>   - Nature: Context-independent, spatially random perturbations
>   - Limitation: Cannot capture semantic, context-dependent errors of real neural networks (Figure 2a and Figure 8)
> - Ours (Adversarial Mask Perturbation):
>   - Method: Adversarial optimization in embedding space guided by model uncertainty
>   - Nature: Context-dependent, semantically meaningful perturbations that concentrate where real models struggle
>   - Advantage: Captures structured, semantic error patterns of real segmentation models
> - Empirical evidence:
>   1.  Direct ablation (Table 4d): When Phoenix uses morphological perturbations instead of adversarial ones, performance drops dramatically, while SegRefiner achieves substantial improvements when equipped with our adversarial perturbations. This bidirectional analysis confirms that our adversarial noise generation drives performance gains
>   2.  Qualitative comparison (Figure 8): Adversarial masks exhibit semantic, contextual errors (boundary confusion, over/under-segmentation in ambiguous regions) that closely match real segmentation failures, while morphological masks show uniform, non-contextual patterns
>   3.  Semantic correlation analysis (Figure 2): Adversarial noise exhibits distribution spanning [-0.6, 0.8] versus morphological noise's narrow [-0.2, 0.3], demonstrating fundamentally different semantic characteristics
>
> **Summary**
> Phoenix is neither adversarial regularization nor traditional augmentation. It repurposes adversarial optimization as a **data generation tool** to synthesize semantically meaningful training examples that capture real-world segmentation error patterns, enabling more effective refinement learning.

---

> ### Author Response · Authors · 2025-11-19
> **Official Comment by Authors (2/2)**
>
> ## W2 & Q2: Unrealistic Distortions and Training Stability
> Phoenix includes multiple mechanisms to ensure realistic perturbations and training stability
>
> 1. IoU Threshold Control (Algorithm 1): The adaptive threshold-guided noise generation ensures perturbations remain within realistic bounds $[\tau, \tau+\epsilon]$. As shown in Figure 4, this parameter directly controls noise intensity; lower thresholds create aggressive noise, higher thresholds produce subtle noise, all remaining realistic.
> 2. Adaptive Step Size with Decay: Algorithm 1 implements automatic step size reduction $(\alpha \leftarrow \alpha/10)$ when perturbations become too aggressive, preventing unrealistic distortions. The process backtracks to the previous state ($E_p^{prev}$) and continues with reduced intensity.
> 3. Diverse Guidance Masks: We randomly select among expansion, contraction, and inversion guides (Section 3.3, lines 250-256), ensuring perturbations don't systematically bias toward unrealistic patterns.
> 4. Empirical Evidence of Stability: Our ablation studies demonstrate robust stability:
>   - Comparison with real model noise (Table 4e): Our adversarial noise achieves 75.7% IoU$^1$, outperforming even masks from real models (67.6% UNet, 65.6% ISNet), demonstrating the perturbations capture realistic and diverse error patterns
>   - Table 8c & Figure 5b: Performance remains stable across wide ranges of $\alpha_0 \in [0.001, 0.1]$ and $N \geq 5$, showing the method is not sensitive to hyperparameters
>
> No filtering mechanism is needed because the adversarial objective inherently optimizes toward semantically meaningful perturbations bounded by our **IoU constraints**. The perturbations cannot become arbitrarily unrealistic as they must: (1) satisfy the IoU threshold, (2) be generated by a frozen pretrained decoder that has learned semantic priors, and (3) operate in embedding space where perturbations naturally align with semantic features.

---

### Official Review · Reviewer_38XS · 2025-10-31

**Soundness:** 2
**Presentation:** 3
**Contribution:** 2
**Rating:** 2
**Confidence:** 3

**Summary:**

This paper introduces a new framework for segmentation mask refinement. The method proposes two primary contributions to generate semantic-aware noisy masks for training and optimize the refiner. The paper's narrative is framed around the AMP component, hypothesizing that this "adversarial noise" better mimics real segmentation errors than the morphological noise used by prior work.

**Strengths:**

1. The core premise of using adversarial attacks not as a test-time failure mode but as a training data generation mechanism is an interesting methodological direction for this problem.
2. The method is benchmarked against SAMRefiner, which also uses the SAM backbone.

**Weaknesses:**

1.  The paper attributes the performance gains to "Learning from Adversity" (AMP), as implied by the title, abstract, and introduction. However, it simultaneously introduces a second, complex, and independent contribution: the CMRL loss. It fails to experimentally disentangle the effects of these two new components, making it impossible to verify the central hypothesis.
2.  The ablation studies are inadequate and confusing. They fail to provide a clear analysis that isolates the individual contributions of AMP and CMRL.
3.  The large performance gap between all SAM-based methods and non-SAM methods suggests the ViT-H backbone itself is a dominant factor. The paper's narrative under-emphasizes this, attributing the improvement primarily to the noise generation strategy.

**Questions:**

1.  To evaluate the paper's central hypothesis, could the authors provide an ablation study that isolates the individual contributions of AMP and CMRL?
2.  Could the authors please clarify the exact settings for the baselines in Tables 4c and 4d? Does the "Morp" setting (Table 4d) use the full CMRL loss?
3.  Could the authors comment on why the paper's narrative (title, abstract) is framed exclusively around the AMP ("Adversity") component?

**Details Of Ethics Concerns:**

N/A.

---

> ### Author Response · Authors · 2025-11-19
> **Official Comment by Authors (1/2)**
>
> We sincerely appreciate the reviewer for the thoughtful and constructive feedback. We particularly appreciate the recognition that repurposing adversarial attacks as a training data generation mechanism represents an interesting methodological direction for mask refinement. Below we address each concern with additional analyses and clarifications.
>
> ---
> ## W1&W2&Q1: Lack of Clear Disentanglement Between AMP and CMRL Contributions
> We acknowledge this is a valid and important concern. While our original submission included ablations for CMRL components (Table 4c) and perturbation methods (Table 4d), we did not provide a clear isolation study comparing: (1) AMP only, (2) CMRL only, and (3) AMP+CMRL together.
> We provide this critical ablation study:
> | Description | Perturbation | CMRL | AP$^{1}$ | AP$^{2}$ | IoU$^{1}$ | IoU$^{2}$ |
> | --- | --- | --- | --- | --- |  --- |  --- |
> | Baseline | Morphological | X | 22.3|45.2 |65.3 |69.9 |
> | AMP only (1st row in Table 4c) | Adversarial | X | 26.9 | 46.1 |71.5 | 74.2 |
> |CMRL only (3rd row in Table 4d) | Morphological | O | 23.8 | 45.5 | 67.8 | 71.0 |
> | Phoenix (AMP+CMRL) | Adversarial | O | 28.7 | 46.9 | 75.7 | 77.1 |
>
> This result demonstrates that
> -   AMP alone provides substantial improvements (+4.6% AP$^1$, +6.2% IoU$^1$), validating our hypothesis that semantic-aware noise is crucial for effective refinement learning.
> -   CMRL alone with morphological noise also improves performance (+1.5% AP$^1$, +2.5% IoU$^1$), demonstrating the value of tri-directional contrastive learning.
> -   AMP+CMRL together achieves the best performance, showing that both components are complementary and contribute meaningfully to the final results (+6.4% AP$^1$, +10.4% IoU$^1$).
> - Synergistic effect: The combined improvement (+10.4% IoU$^1$) exceeds the sum of individual contributions (+6.2% + 2.5% = +8.7% IoU$^1$), demonstrating true synergy. This occurs because CMRL is more effective when encountering realistic noise patterns from AMP; the semantically meaningful error distributions enable CMRL to construct more informative tri-directional contrastive relationships between ground truth, noisy input, and predictions, leading to superior refinement learning.
>
> We have added this result to Table 4f in the main manuscript, along with the analysis in the Section 4.4 "Effect of Individual Components".
>
> ---
> ## Q2: Confusing Ablation Studies
> We apologize for the confusion in our original presentation. We now clarify the experimental settings:
>
> - Table 4c (CMRL components): All experiments use adversarial perturbation (AMP). The baseline (1st row: all components = x) uses only pixel-wise loss with AMP, which corresponds to "AMP only" in the table above (26.9% AP$^1$, 46.1% AP$^2$).
> - Table 4d (Perturbation methods): The 3rd row (Phoenix + Morp) uses the morphological noise and CMRL loss, which corresponds to "CMRL only" in the table above (67.8% IoU$^1$, 71.0% IoU$^2$).

---

> ### Author Response · Authors · 2025-11-19
> **Official Comment by Authors (2/2)**
>
> ## W3: Backbone Effect
> We appreciate the reviewer raising this important consideration regarding the backbone's contribution. We provide two key pieces of evidence demonstrating that the ViT-H backbone itself is not the dominant factor.
>
> **Evidence 1.** Table 4d provides compelling evidence:
>
> | Method | Perturbation | IoU$^1$ | IoU $^2$ |
> | --- | --- | --- | --- |
> | SegRefiner (Diffusion-UNet) | Morphological | 58.7 | 74.2 |
> | **SegRefiner (Diffusion-UNet)** | **Adversarial** | **71.8** | **76.6** |
> | **Phoenix (ViT-H)** | **Morphological** | **67.8** | **71.0** |
> | Phoenix (ViT-H) | Adversarial | 75.7 | 77.1 |
>
> SegRefiner with our adversarial perturbation outperforms Phoenix with ViT-H when Phoenix uses morphological noise. This directly demonstrates that the noise generation strategy (AMP) contributes more than the backbone architecture itself.
>
> **Evidence 2.** Phoenix with smaller backbone outperforms SAMRefiner with ViT-H
> -   Phoenix + ViT-B: 22.4% AP$^1$ (Table 8f)
> -   SAMRefiner + ViT-H: 21.8% AP$^1$ (Table 1a)
>
> Our ViT-B variant outperforms SAMRefiner + ViT-H (+0.6% AP$^1$), demonstrating that our technical contributions (AMP + CMRL) provide value independent of backbone capacity.
>
> ---
> ## Q3: Narrative Framing
> We sincerely appreciate the reviewer's thoughtful feedback on our paper's presentation. We acknowledge that our framing may have overemphasized the AMP contribution compared to CMRL. We would like to respectfully clarify our framing choices and address this concern.
>
> - Rationale for title emphasis on AMP: The title reflects AMP's role as the conceptual foundation that enables effective refinement learning. CMRL naturally complements this foundation; it is not an auxiliary addition but rather the mechanism that fully realizes the potential of semantic-aware noise patterns through explicit relationship modeling. Table 4f demonstrates this complementary nature: both components provide substantial individual gains, and their combination yields synergistic improvements.
> - Evidence of integrated design: The tight coupling between AMP and CMRL is evident in our framework (Figure 3). AMP generates semantically meaningful error patterns that expose the model to realistic failure modes, while CMRL learns to correct these patterns by modeling tri-directional relationships. This integration is fundamental to Phoenix's design.
> - Current balance in the manuscript: Our abstract explicitly presents both AMP and CMRL as "two key innovations" with equal structural weight, and the introduction provides detailed motivation and technical description for both components. Section 3 dedicates equal space to each (3.3 for AMP, 3.4 for CMRL), and our experimental results consistently evaluate both contributions.
>
> We believe the current framing accurately represents our contributions, with the title emphasizing the novel conceptual direction while the body text provides balanced, detailed treatment of both innovations. We are happy to make specific adjustments if the reviewer identifies particular sentences that would benefit from rebalancing, while ensuring both contributions receive appropriate emphasis.

---

### Official Review · Reviewer_Nrbx · 2025-11-03

**Soundness:** 3
**Presentation:** 3
**Contribution:** 3
**Rating:** 6
**Confidence:** 4

**Summary:**

The paper introduces Phoenix, a framework for mask refinement in image segmentation, aiming to address persistent issues in mask boundaries, semantic consistency, and structural integrity. Phoenix augments the current state-of-the-art by leveraging adversarial perturbation in the embedding space (Adversarial Mask Perturbation, AMP) to generate semantically-plausible noise, and a tri-directional Contrastive Mask Refinement Learning (CMRL) loss to refine noisy masks. The approach is built atop the Segment Anything Model (SAM) and is extensively evaluated—outperforming strong baselines in semi-supervised, weakly-supervised, fine-grained, and zero-shot transfer settings across multiple datasets.

**Strengths:**

**Innovation in Noise Modeling:** The introduction of AMP, which generates mask perturbations via adversarial embedding-level attacks rather than naive morphological operations, provides a more challenging and semantically aligned training regime. As shown in Figure 2, adversarial noise patterns demonstrate a higher semantic correlation with segmentation errors compared to morphological noise.

**Powerful Tri-Directional Contrastive Loss:** The CMRL loss explicitly models relationships between ground-truth, noisy, and refined masks, encouraging foreground–background separation, intra-class consistency, and self-improvement. The design is mathematically well-motivated—Section 3.4 details feature-space losses that move beyond basic pixel-level objectives.

**Comprehensive Empirical Evidence:** The paper presents broad quantitative comparisons (see Tables 1, 2, and 3), consistently demonstrating strong improvements over state-of-the-art refinement strategies, especially SAMRefiner and SegRefiner. The results generalize across full-supervision, semi-/weak-supervision, and domain transfer tasks.

**Weaknesses:**

**Theoretical Clarity of AMP Mechanism:** While the AMP methodology is generally well-explained (see Algorithm 1), the theoretical analysis connecting adversarial perturbation in embedding space to task-specific semantic error diversity is somewhat hand-wavy. The justification for why embedding-level attacks yield realistic error patterns is primarily based on empirical distributions (see Figure 2c), and the claimed proportionality between gradient norm and local uncertainty relies heavily on assumed model properties. No ablation examines weaknesses of AMP, e.g., failure modes if embedding space is not predictive of real error patterns.

**Contrastive Loss Formulation Details:** The mathematical construction of the tri-directional loss in Section 3.4 (multiple feature regions derived from mask overlaps) is intricate but under-explained for readers unfamiliar with the domain. The paper would benefit from explicit algorithmic pseudo-code or stepwise explanation of how region masks and projection features are computed and batched in a practical setting. Furthermore, the mapping from features to projected space (projector $g$) is presented as an afterthought, with no architectural or optimization ablation provided.

**Ambiguities and Reproducibility Gaps:** Key implementation choices for both guiding mask selection and the adversarial update process are reported as “randomly chosen” or based on heuristics (Section 3.3 and Appendix B). For instance, Table 4d (mask perturbation method) and Table 8d (guidance mask ablation) demonstrate sensitivity to mask type, but practical selection protocols are not systematically explored. Minor but notable: several equations (especially for the loss in Section 3.4) lack definitive bounds for summations, and some index notations could be clarified for reproducibility.

**Questions:**

Please see Weaknesses.

---

> ### Author Response · Authors · 2025-11-19
> **Official Comment by Authors (1/3)**
>
> We sincerely thank Reviewer Nrbx for the thoughtful and constructive feedback. We are encouraged by your recognition of our innovation in noise modeling, the tri-directional contrastive framework, and comprehensive empirical validation. Below, we address each concern.
>
> ---
> ## W1: Theoretical Clarity of AMP Mechanism
> We appreciate this important feedback and provide both theoretical justification and additional empirical analysis.
>
> **Theoretical Foundation:** The connection between embedding-level gradients and semantic uncertainty is grounded in established uncertainty quantification theory. As detailed in our citation to *Kendall & Gal (2017)*, the gradient magnitude in neural network output space relates inversely to prediction confidence. Specifically, for a fixed decoder with parameters $\theta_{dec}$, the gradient $\|\nabla_{E_p} f_{dec}(E_{img}, [E_p; E_v]; \theta_{dec})\|$ has higher magnitude in regions where the model's probability distribution $p(y|E_{img}, [E_p; E_v]; \theta_{dec})$ exhibits lower confidence.
> Our FGSM-based perturbation amplifies errors precisely in these uncertain regions because:
> 1.  The $\text{sign}(\nabla_{E_p} \mathcal{L}_{adv})$ update step moves embeddings along the steepest descent direction of adversarial loss
> 2.  High-gradient regions correspond to decision boundaries where small perturbations cause large output changes
> 3.  These boundaries naturally align with semantic edges and ambiguous regions where real segmentation models struggle
>
> **Why Embedding-Level vs Pixel-Level Perturbation?** Our choice of embedding-level attacks over traditional pixel-level adversarial attacks is deliberate and offers key advantages:
> 1.  Semantic Perturbations: As noted in our citation to *Huang & Zhang (2020)*, embedding-space attacks operate at a higher semantic level, directly manipulating learned feature representations rather than low-level pixel values. This naturally produces semantically meaningful errors (e.g., boundary confusion, object merging) rather than imperceptible pixel noise.
> 2.  Computational Efficiency: Operating on decoder embeddings (4M parameters, 6ms per update) is 100× faster than backpropagating through the full encoder (637M parameters). We reuse frozen image embeddings from a single encoder forward pass, making AMP highly efficient during training.
> 3.  Task Alignment: Pixel-level attacks would require perturbing the input image, which doesn't directly simulate mask errors. Our approach perturbs the mask representation pathway, directly modeling the types of errors segmentation models make.
>
> **Empirical Validation of AMP Effectiveness:** We emphasize that our paper provides strong evidence that AMP successfully generates realistic error patterns:
> -   Table 4d: Direct comparison shows adversarial perturbation is critical; replacing it with morphological noise drops performance dramatically (75.7 $\rightarrow$ 67.8% IoU$^1$ and 77.1 $\rightarrow$ 71.0% IoU$^2$for Phoenix).
> -   Table 4e: AMP generates more diverse and challenging patterns than real model errors. Training with adversarial noise (75.7 IoU$^1$) outperforms training on real model noises (67.6 and 65.6 IoU$^1$) by 8-10%. This demonstrates AMP captures broader error distributions than individual real models.
>
> **Systematic Control via IoU Threshold:** Importantly, AMP doesn't produce arbitrary or uncontrolled noise; the IoU threshold $\tau$ provides systematic control over perturbation magnitude:
> -   Algorithm 1 ensures generated masks satisfy $IoU \in [\tau, \tau+\epsilon]$ through adaptive step size adjustment
> -   Figure 4 demonstrates controllable noise intensity across $\tau \in [0.5, 0.9]$
> -   Table 8a shows robust performance across different $\tau$ ranges, with $\mathcal{U}(0.3, 0.9)$ achieving optimal diversity

---

> ### Author Response · Authors · 2025-11-19
> **Official Comment by Authors (2/3)**
>
> ## W2: Contrastive Loss Formulation Details
>
> Following your valuable feedback, we have added a new subsection titled "Implementation Details of Contrastive Mask Refinement Learning (CMRL)" in Appendix B.6. This subsection provides:
>
> 1.  Step-by-step explanation of six region mask computation through logical operations
> 2.  Feature projection and sampling strategy for computational efficiency
> 3.  Detailed description of each loss component's role and mechanism
> 4.  Projector architecture specification and ablation study
> 5.  PyTorch-style pseudo-code (Algorithm 2) for immediate reproducibility
>
> ~~~
> # Input: image features F (Bx32x256x256),
> # masks M_t (GT), M_n (Noisy Mask), M_r (Refined Mask) (Bx1xHxW)
>
> # Step 1: Project and normalize features
> projector = MLP_layer(in_channels=32, out_channels=32, num_layers=3)
> P = F.normalize(projector(F), p=2, dim=1)  # (Bx32x256x256)
>
> # Step 2: Define six region masks (binarize and detach)
> M_t_bin = (M_t > 0.5).float().detach()
> M_n_bin = (M_n > 0.5).float().detach()
> M_r_bin = (M_r > 0.5).float().detach()
>
> T_fg = (M_t_bin==1) & (M_n_bin==1) & (M_r_bin==1) # True positive
> T_bg = (M_t_bin==0) & (M_n_bin==0) & (M_r_bin==0) # True negative
> S_fg = (M_t_bin==1) & (M_n_bin==0) & (M_r_bin==1) # Success FN->TP
> S_bg = (M_t_bin==0) & (M_n_bin==1) & (M_r_bin==0) # Success FP->TN
> F_fg = (M_t_bin==1) & (M_n_bin==0) & (M_r_bin==0) # Failure: uncorrected FN
> F_bg = (M_t_bin==0) & (M_n_bin==1) & (M_r_bin==1) # Failure: uncorrected FP
>
> # Step 3: Compute losses per batch item
> for b in range(B):
>   feat = P[b].view(C, -1).T  # (HxW, C) - flattened normalized features
>
>   # Sample pixels from each region (max 256 samples per region)
>   anchor_fg = sample_pixels(feat, F_fg[b], num=256)  # Failure foreground
>   anchor_bg = sample_pixels(feat, F_bg[b], num=256)  # Failure background
>   pos_fg = sample_pixels(feat, T_fg[b] | S_fg[b], num=256) #Correct foreground
>   pos_bg = sample_pixels(feat, T_bg[b] | S_bg[b], num=256) #Correct background
>
>   # Intra-class: Pull failures toward correct same-class features (InfoNCE)
>   # For foreground: anchor_fg -> pos_fg
>   sim_pos = (anchor_fg @ pos_fg.T) / tau  # (NxM)
>   sim_all = (anchor_fg @ feat.T) / tau     # (NxHxW)
>   L_intra_fg = -mean(logsumexp(sim_pos, dim=1) - logsumexp(sim_all, dim=1))
>   # Similarly for background: L_intra_bg
>
>   # Inter-class: Push failures away from opposite-class features
>   # For foreground failures -> background regions
>   sim_opposite = (anchor_fg @ pos_bg.T) / tau
>   L_inter_fg = mean(log(1 + exp(sim_opposite).sum(dim=1)))
>   # Similarly for background: L_inter_bg
>
>   # Self-improvement: Guide failures toward success regions
>   success_feat = sample_pixels(feat, S_fg[b] | S_bg[b], num=512)
>   failure_feat = sample_pixels(feat, F_fg[b] | F_bg[b], num=512)
>   sim_success = (failure_feat @ success_feat.T) / tau
>   sim_all = (failure_feat @ feat.T) / tau
>   L_self = -mean(logsumexp(sim_success, dim=1) - logsumexp(sim_all, dim=1))
>
> # Step 4: Combine losses
> L_CMRL = 0.4*L_intra + 0.4*L_inter + 0.2*L_self
> ~~~
>
> For projector architecture and ablation, the projector $g$ consists of three 1×1 Conv2d layers with LayerNorm and GELU activations between layers. Our ablation study demonstrates that the 3-layer MLP achieves optimal performance:
> | Projector | AP$^1$ | AP$^2$ |
> | --- | --- | --- |
> | Identity (No projection) | 27.1 | 46.2 |
> | 1-layer MLP | 27.9 | 46.5 |
> | 2-layer MLP | 28.5 | 46.7 |
> | 3-layer MLP (ours) | 28.7 | 46.9 |
> | 4-layer MLP | 28.6 | 46.8 |

---

> ### Author Response · Authors · 2025-11-19
> **Official Comment by Authors (3/3)**
>
> ## W3: Ambiguities and Reproducibility Gaps
> We acknowledge these concerns and provide clear protocols for all design choices:
> 1.  Practical Protocol for Guidance Masks (Section 3.3, Table 8d): We employ a simple random sampling strategy where each of the three guidance types (i.e., expansion, contraction, and inversion) is selected with probability 1/3 during training. This approach is both straightforward to implement and empirically optimal. Table 8d shows that using all three types (28.7% AP$^1$) substantially outperforms using only expansion (26.9%) or contraction (27.1%) or any combination of two (28.5%). Our recommendation for practitioners is using the random mixture of all three guidance types to ensure robust refinement performance across diverse error patterns.
> 2.  IoU Threshold $\tau$ Selection Strategy (Algorithm 1, Table 4a and Table 8c): For each noisy mask generation, we sample $\tau$ uniformly from [0.3, 0.9]. This range is carefully chosen based on Table 4a and Table 8c, which demonstrates that the full range outperforms narrower alternatives by providing exposure to both challenging ($\tau$=0.3) and moderate ($\tau$=0.9) noise patterns. Algorithm 1 ensures systematic noise control through iterative refinement with adaptive step sizing, guaranteeing $IoU \in [\tau, \tau+\epsilon]$ for every generated mask, eliminating concerns about arbitrary or uncontrolled perturbations.
>
> We will provide complete code, pretrained models, and step-by-step implementation guidelines upon acceptance.
>
> ---
> ## Equation Notation Improvements in Section 3.4
> We acknowledge the notation ambiguities and will make the following revisions to Section 3.4:
> 1. Add notation block: Let $F\in \mathbb{R}^{C\times H\times W}$ denote upsampled image embeddings. We apply projector $g$ to obtain $P=g(F)\in\mathbb{R}^{C\times H\times W}$. For each position $i\in\Omega=\{1, ..., H\times W\}, p_i\in\mathbb{R}^c$, denotes the feature vector at position $i$. The expectation $E_{i\in R}[\cdot]$ represents uniform sampling, and the similarity is $sim(p_i, p_j) = p_i^T p_j$ (cosine similarity on L2-normalized features).
> 2. Clarify summation bounds in Equations (1) and (3): Replace ambiguous $\sum_{k}$ with explicit $\sum_{k\in\Omega}$ where $\Omega=\{1, ..., H\times W\}$ denotes all spatial positions

---

> > ### Comment · Reviewer_Nrbx · 2025-11-19
> > **Thanks for Rebutaal**
> >
> > All concerns are addressed, and I decide to raise my score and accept this paper.

---

### Author Response · Authors · 2025-11-19
**Common Response**

We sincerely appreciate all reviewers for their thoughtful and constructive feedback on our work. We are grateful for the time and effort invested in providing detailed evaluations, which have been invaluable in strengthening our paper. The insightful suggestions and critical observations have helped us clarify our contributions and address important concerns.

---
**Summary of Our Contributions.** Phoenix introduces a novel framework for mask refinement that addresses the fundamental limitation of existing approaches, $i.e.,$ their reliance on simplistic morphological noise that fails to capture real segmentation errors. Our two key innovations are: (1) **Adversarial Mask Perturbation (AMP)**, which generates semantically meaningful, contextually aware noise patterns through adversarial embedding attacks, providing controllable yet realistic training data, and (2) **Contrastive Mask Refinement Learning (CMRL)**, a tri-directional contrastive framework that explicitly models relationships between ground truth, noisy input, and refined predictions to learn better refinement patterns. Our extensive experiments demonstrate significant improvements across diverse segmentation tasks.
Moreover, Phoenix offers significant practical impact and broad applicability: it serves as a **plug-and-play module** that enhances diverse segmentation models without architectural changes (Tables 1-3, 6a), enables dataset annotation refinement for quality improvement (Table 6b), and demonstrates zero-shot generalization to medical imaging and urban scenes (Tables 7a-b).

---
**Strengths Acknowledged by Reviewers.** We are encouraged that reviewers recognized several strengths of our work:
- Innovation in noise modeling and AMP methodology (Reviewers Nrbx, 38XS, ADPo, bEY1): The adversarial approach to noise generation was acknowledged as an interesting and technically sound contribution that provides more challenging and semantically aligned training data.
- Powerful contrastive learning framework (Reviewer Nrbx): The tri-directional CMRL design was recognized as mathematically well-motivated and moving beyond basic pixel-level objectives.
- Comprehensive and strong empirical results (Reviewers Nrbx, bEY1): Our systematic evaluation across multiple settings, including semi-supervised, weakly-supervised, fine-grained segmentation, and zero-shot generalization, demonstrates consistent improvements over state-of-the-art methods.
- Thorough experimental analysis (Reviewers ADPo, bEY1): The extensive ablation studies, efficiency analyses, and additional applications in the appendix were appreciated for providing thoughtful experimentation.
- Clear presentation and well-motivated method (Reviewer bEY1): The method description, implementation details, and code availability commitment were valued.

---
**Addressing Reviewer Concerns.** We have carefully considered all concerns raised by the reviewers and provide detailed responses to each reviewer individually below. Based on the valuable feedback, we have **updated the paper accordingly (indicated in blue text in the revised manuscript)**.

---
We remain open to further discussion and feedback during the rebuttal period. We believe the additional experiments and clarifications significantly strengthen the paper and address the core concerns raised by all reviewers.

---

### Meta-Review · Area_Chair_A55A · 2025-12-22

**Summary:**

The reviews for this paper are mixed. This paper was reviewed by four experts in the field and received a wide range of scores, spanning Reject (2), Marginal Reject (4), Marginal Accept (6), and Accept (8).

This paper introduces Phoenix, a semantic-aware technique for segmentation mask refinement that targets persistent issues such as inaccurate boundaries, semantic inconsistencies, and structural artifacts in modern segmentation models. Phoenix is a incremental contribution based on SAM  that rely on adversarial noise introduce by Adversarial Mask Perturbation (AMP) to generate semantically plausible noise that better mimics real segmentation error. Also Phoenix uses  Contrastive Mask Refinement Learning (CMRL), a tri-directional contrastive objective that aligns the ground-truth, the noisy, and the  refined masks while preserving intra-class consistency and inter-class separation. Extensive experiments across semi-supervised, weakly supervised, fine-grained, and zero-shot transfer settings demonstrate that Phoenix consistently outperforms strong baseline methods on multiple datasets.

While several reviewers acknowledge the paper’s good empirical performance and interesting ideas, there is no clear consensus in favor of acceptance. Hence, based on the reviews, I side with the reviewers recommending rejection. The authors are encouraged to consider the reviewers’ comments, to improve the paper for submission elsewhere.

**Reviewer Concerns:**

The reviewers initialy raised the following main concerns:
1. Insufficient disentanglement of contributions: Several reviewers raise a key issue, namely that the article presents two important elements, namely antagonistic mask perturbation (AMP) and contrastive learning loss through mask refinement (CMRL), without convincingly isolating their individual effects. The ablation studies are deemed insufficiently clear to validate the article's central hypothesis that AMP is the main driver of the observed improvements.
2. Attribution of gains and backbone dominance: the reviewers noted that much of the performance improvement could be attributed to the use of a robust SAM (ViT-H) base structure rather than the proposed improvement strategy itself. The article emphasizes AMP as the main source of improvement, but does not sufficiently contextualize or control for the impact of the underlying base structure in comparisons with non-SAM-based methods.
3. Conceptual and theoretical clarity: While the method is technically sound, reviewers expressed concerns about the theoretical justification of AMP. The connection between embedding-space adversarial perturbations and realistic segmentation error distributions is argued largely empirically, with limited formal grounding. Similarly, the CMRL formulation, though powerful, is complex and under-explained, making it difficult to fuly assess or reproduce.
4. Although the method appears technically correct, reviewers expressed concerns about the theoretical justification for AMP. The link between adversarial perturbations of the integration space and realistic distributions of segmentation errors is largely argued empirically, with limited formal and theoretical explanations. Similarly, the CMRL formulation, while powerful, is complex and insufficiently explained, making it difficult to fuly evaluate or reproduce. As an AC that read the final version I totally agree with that point.
5. Reproducibility and presentation issues: Several implementation details are based on heuristic or “random” choices without systematic justification. Important aspects of loss formulation, mask selection, and antagonist update procedure lack clarity, and certain presentation issues (notation, figure order, missing details) further hinder reproducibility.

Based on the reviews, I side with the reviewers recommending not to accept the paper. While the work demonstrates strong empirical results and addresses an important problem, the lack of clear experimental disentanglement of contributions, concerns about attribution of gains, and gaps in theoretical and methodological clarity prevent the paper from meeting the bar for acceptance at ICLR in its current form.

**Reviewer Scores:**

If there had been no issue of the Review period of the ICLR this year, I think the final score would have been 8, 8, 2, and 4, which would have made this article borderline. I spent a lot of time reading the rebuttal and the article. I must admit that I agree with the reviewers on the issue of presentation. Response to **W1: Theoretical Clarity of AMP Mechanism** is not clear to me and does not justify anything. Worse still, it is not clear why the authors claim that uncertainty is related to an adversarial attack. Can they prove this mathematically? Regarding weakness **W2: Contrastive Loss Formulation Details**, I do not think the appendix is the appropriate place for such a crucial section. Furthermore, this section is important and a part should be added where the authors explain how to choose the parameters $\lambda_{\mbox{intra}}$, $\lambda_{\mbox{inter}}$ and $\lambda_{\mbox{self}}$. Without this, it is impossible to use such an incremental article. Regarding the weakness: **W3: Backbone Effect** : I think the authors should have tested and compared more backbones, not just two. It is therefore difficult to know whether the claim is truly generalizable. But the answer to **W1&W2&Q1: Lack of Clear Disentanglement Between AMP and CMRL Contributions** seems to answer the question.
It is important to note that question **Q3: Narrative Framing** does not seem to address the issue, because after reading the new version of the article, I share exactly the same opinion as the reviewer. This means that the author has not taken the reviewer's mandatory recommendation into account. Unfortunately, without taking this recommendation into account, the article resembles a set of many different Lego pieces that are set together to build a new structure with no meaning other than to have the best possible structure for publication…. Articles submitted to ICLR should be more than just a Lego structure. The answer to **W1 & Q1: Difference between augmentation strategies and AMP** is not clear to me, and it would have been great to formalize it mathematically to prove that the proposed approach is not a copy-paste of antagonist training.


All these points mean that, based on all the responses and the fact that there has been virtually no interaction on this article, I prefer to reject the article as it stands. I recognize that the article is interesting, but it needs to be rewritten, with better justification of the difference between the architecture and previous work and better justification of the design of the proposed solution.

---

### Decision · Program_Chairs · 2026-01-26

Reject